# OFFLINE CLUSTERING OF LINEAR BANDITS:
# THE POWER OF CLUSTERS UNDER LIMITED DATA

## ABSTRACT

Contextual multi-armed bandit is a fundamental learning framework for making a sequence of decisions, e.g., advertising recommendations for a sequence of arriving users. Recent works have shown that clustering these users based on the similarity of their learned preferences can accelerate the learning. However, prior work has primarily focused on the online setting, which requires continually collecting user data, ignoring the offline data widely available in many applications. To tackle these limitations, we study the offline clustering of bandits (Off-ClusBand) problem, which studies how to use the offline dataset to learn cluster properties and improve decision-making. The key challenge in Off-ClusBand arises from data insufficiency for users: unlike the online case where we continually learn from online data, in the offline case, we have a fixed, limited dataset to work from and thus must determine whether we have enough data to confidently cluster users together. To address this challenge, we propose two algorithms: Off-$C^2$LUB, which we show analytically and experimentally outperforms existing methods under limited offline user data, and Off-CLUB, which may incur bias when data is sparse but performs well and nearly matches the lower bound when data is sufficient. We experimentally validate these results on both real and synthetic datasets.

## 1 INTRODUCTION

Contextual bandits (Chu et al., 2011; Agrawal & Devanur, 2016) extend the fundamental multi-armed bandit (MAB) framework (Robbins, 1952; Lai & Robbins, 1985; Auer et al., 2002) for optimizing online decision-making to settings where rewards depend on given contexts, often through a linear function. Recent works have proposed to accelerate the learning in contextual MAB by taking advantage of the *clustered* structure in many real-world applications, e.g., social networking (Delporte et al., 2013; Bitaghsir et al., 2019) and advertisement recommendation (Yan et al., 2022; Aramayo et al., 2023). This framework assumes that a given set of users arrives at the system in sequence; users generally arrive multiple times within the sequence (e.g., users visiting a website multiple times and being shown an ad at each visit). These users can be partitioned into clusters that share the same or similar preferences. The learning agent adaptively clusters users into groups based on their observed responses to the actions taken, leveraging the preference similarities within these groups to improve its decisions. This online clustering approach enables more efficient and personalized decision-making, which has been extensively studied across diverse settings (Gentile et al., 2014; Li et al., 2019; Ban et al., 2024). Detailed related works can be found in Appendix A.

Despite the significant progress in clustering-based bandit frameworks, prior work has largely overlooked scenarios where *offline data* plays a crucial role in learning. For example, advertisement platforms often accumulate extensive historical data on the effectiveness of given ads across different users. This data can be used to pre-compute user clusters, enabling more informed and efficient future recommendations. MAB approaches have also been applied to choosing medical treatments (Komorowski et al., 2018; Zhou et al., 2019), for which patients with similar physiological profiles often exhibit comparable responses. However, ethical and practical constraints make it infeasible to directly experiment with treatments on patients. Instead, learning must be conducted using preexisting historical treatment data. To address this gap, we introduce, to the best of our knowledge, the *first algorithms for offline clustered bandits*. Our approach leverages offline data to identify clusters and optimize decision-making in settings where online exploration is constrained or infeasible, paving the way for broader applications of MAB-based decision-making.

Extending bandit clustering algorithms from the online to the offline setting poses a **fundamentally new challenge**: unlike in the online setting, where data continually arrives during algorithm execution, the offline setting relies on a *fixed offline dataset*. The continual arrival of new data in the online setting allows the algorithm to iteratively refine its understanding of the clustering structure; indeed, over $T$ rounds, only $O(\log(T))$ rounds are needed to accurately identify clusters. However, in the offline setting, the number of samples per user is inherently limited. For instance, some users may have sparse historical data due to infrequent activity, or data on certain treatments or actions may be scarce. This limitation creates significant challenges in accurately estimating the preferences of users with limited data and thus in clustering users with similar preferences. We thus develop criteria to confidently identify similar users, and then account for potential clustering errors—caused by limited or noisy data—in algorithm design and analysis. Unlike standard offline statistical learning (Zhang & Zhang, 2014; Cai & Guo, 2017; Duan & Wang, 2023), which focuses on parameter estimation, our setting also requires *structural decisions*, including grouping users and action selections. Moreover, multi-task statistical learning (Duan & Wang, 2023) considers related tasks, but it does not address the challenge of sparse and different amounts of heterogeneous data among users. In contrast, our algorithms leverage these differences to distinguish users and guide clustering.

To address this challenge, we introduce the *Offline Clustering of Bandits* (Off-ClusBand) problem. In Off-ClusBand, there are $U$ users represented by the set $\mathcal{U}$, each associated with a preference vector. The users are partitioned into $J$ clusters, with all users in a cluster sharing the same preference vector (Gentile et al., 2014; Li & Zhang, 2018; Li et al., 2019; Wang et al., 2025; Li et al., 2025). However, the partition, the number of clusters, and the preference vectors themselves are unknown to the algorithm. An input dataset $\mathcal{D}$ is provided, containing historical samples of the actions taken and rewards received for each user, assumed to be a linear function of user actions and preference vectors. The objective is to identify users with similar preferences to aggregate their offline data, and design an algorithm that minimizes the suboptimality gap—the difference between the reward achieved by the algorithm and the optimal reward achievable in hindsight—for any given test user $u_{\text{test}} \in \mathcal{U}$ and an arbitrary action set $\mathcal{A}_{\text{test}}$, which may not have been observed in the offline dataset.

**Contributions.** (i) In Section 2, we formalize the Off-ClusBand problem. We then design the Off-C$^2$LUB algorithm (Algorithm 1) in Section 3, which addresses the challenge of inaccurate clustering due to limited user data by introducing a parameter $\hat{\gamma}$ that thresholds similar users. In

Table 1: Recommended Algorithms in Different Scenarios ($\gamma$ is the minimum gap between users in different clusters).

| $\gamma$ 
 Dataset | Known | Unknown |
|---|---|---|
| **Sufficient** | Equivalent | Off-CLUB |
| **Insufficient** | Off-C$^2$LUB | Off-C$^2$LUB |

Theorem 3.4, we upper bound Off-C$^2$LUB's suboptimality gap, accounting for two types of error. The *noise* decreases as more users' samples are aggregated, and this reduction accelerates with an increase in $\hat{\gamma}$. Conversely, the *bias*, which comes from aggregating data from users with distinct preferences, may increase with $\hat{\gamma}$. By selecting $\hat{\gamma} \leq \gamma$, where $\gamma$ is the minimum gap between any two users from different clusters, the algorithm incurs zero bias but higher noise. We analyze the influence of $\hat{\gamma}$ and provide policies for selecting it under different scenarios, nearly matching the lower bound of the suboptimality gap that we derive in Section 4.3.

(ii) We then present the Off-CLUB algorithm (Algorithm 2) in Section 4, which does not require $\hat{\gamma}$. Assuming sufficient user data for accurate clustering, we upper bound Off-CLUB's suboptimality in Theorem 4.2. However, Off-CLUB struggles when the dataset is sparse or lacks sufficient samples. The recommended algorithms based on different data conditions is provided in Table 1.

(iii) Finally, in Section 5, we validate our algorithms through synthetic and real-world experiments. These experiments demonstrate the strong performance of Off-C$^2$LUB with different amounts of data, especially with our proposed $\hat{\gamma}$-selection policies. Off-C$^2$LUB reduces the suboptimality gap by over 60% compared to traditional clustering algorithms. Moreover, they confirm the effectiveness of Off-CLUB under sufficient data and highlight its limitations when data is insufficient.

## 2 PROBLEM FORMULATION

In this section, we introduce the setting for the "Offline Clustering of Bandits (Off-ClusBand)" problem. We consider $U$ *users*, represented as $\mathcal{U} = \{1, \cdots, U\}$, where each user $u$ is associated with

a *preference vector* $\boldsymbol{\theta}_u$. To capture the heterogeneity within the user population, users are grouped into $J(\leq U)$ disjoint *clusters*, with all users in the same cluster sharing an identical preference vector. Precisely, we define $\mathcal{U} = \bigcup_{j=1}^{J} \mathcal{V}(j)$, where $\mathcal{V}(j)$ denotes the set of users within cluster $j$, hence $\mathcal{V}(j) \cap \mathcal{V}(j') = \emptyset$ for $j \neq j'$. Additionally, $\boldsymbol{\theta}_u = \boldsymbol{\theta}_{u'}$ if and only if there exists a $j \in [J]$ such that $\{u, u'\} \subseteq \mathcal{V}(j)$. The cluster to which user $u$ belongs is denoted by $j_u$, and the shared preference vector for users in cluster $\mathcal{V}(j)$ is denoted as $\boldsymbol{\theta}^j$. Notably, both the actual partition of clusters and the number of clusters remain unknown to the algorithm. For user $u$, we refer to users in the same cluster as *homogeneous users*, and those in different clusters as *heterogeneous users*.

In the offline setting, we are given a dataset $\mathcal{D} = \{\mathcal{D}_1, \cdots, \mathcal{D}_U\}$ comprising offline samples collected from $U$ users. For each user $u$, we observe $N_u$ samples, forming $\mathcal{D}_u = \{(\boldsymbol{a}_u^i, r_u^i)\}_{i=1}^{N_u}$, where $\{\boldsymbol{a}_u^i\}_{i=1}^{N_u}$ represents the $N_u$ given actions, and $\{r_u^i\}_{i=1}^{N_u}$ denotes the corresponding rewards, independently drawn as $r_u^i \sim R(\boldsymbol{a}_u^i)$. Here, $R(\boldsymbol{a}_u^i)$ is a 1-subgaussian reward distribution associated with the action $\boldsymbol{a}_u^i$, and $\mathbb{E}[r_u^i] = \langle \boldsymbol{\theta}_u, \boldsymbol{a}_u^i \rangle$ represents the mean reward for user $u$ under action $\boldsymbol{a}_u^i$. We also denote the total number of samples for a set of users $\mathcal{S}$ by $N_{\mathcal{S}} = \sum_{v \in \mathcal{S}} N_v$. Additionally, we make the following standard action regularity assumption common in clustering of bandits literature (Gentile et al., 2014; Wang et al., 2023a; Dai et al., 2024a; Wang et al., 2025).

**Assumption 2.1** (Action Regularity). Let $\rho$ be a distribution over $\{\boldsymbol{a} \in \mathbb{R}^d : \|\boldsymbol{a}\|_2 \leq 1\}$ whose covariance matrix $\mathbb{E}_{\boldsymbol{a} \sim \rho}[\boldsymbol{a}\boldsymbol{a}^\top]$ is full rank with minimum eigenvalue $\lambda_a > 0$. For any fixed unit vector $\boldsymbol{\theta} \in \mathbb{R}^d$, the random variable $(\boldsymbol{\theta}^\top \boldsymbol{a})^2$, with $\boldsymbol{a} \sim \rho$, has sub-Gaussian tails with variance upper bounded by $\sigma^2$. For each dataset $\mathcal{D}_u$, each action $\boldsymbol{a}_u^i$ is selected from a finite candidate set $\mathcal{S}_u^i$ with size $|\mathcal{S}_u^i| \leq S$ for any $i \in [N_u]$, where the actions in $\mathcal{S}_u^i$ are independently drawn from $\rho$.

Moreover, we assume the *smoothed regularity parameter* $\tilde{\lambda}_a = \int_0^{\lambda_a} \left(1 - e^{-\frac{(\lambda_a - x)^2}{2\sigma^2}}\right)^S \mathrm{d}x$ is known to the algorithm.

*Remark* 2.2 (Discussion on Assumption 2.1). Assumption 2.1, adapted from Wang et al. (2023a) and widely used in recent clustering of bandits studies (Dai et al., 2024a; Wang et al., 2025), ensures sufficient coverage of the action space so that information is available along every dimension of the preference vector—a key requirement for distinguishing users across clusters. This assumption is particularly suitable when offline actions are drawn from a finite action space with size bounded by $S$ (e.g. in datasets generated by online logging policies with finite action sets per round). Experiments in Section 5 and Appendix F further validate the effectiveness of our algorithms on offline data generated under different distributions and logging policies. For completeness, we defer discussion of alternative forms of the regularity assumption, such as cases where offline samples are drawn independently from an infinite distribution $\rho$, to Appendix B.

Let $\pi$ denote an algorithm that selects an action from an action set $\mathcal{A}_{\text{test}} \subseteq \mathbb{R}^d$ for any given user $u_{\text{test}} \in \mathcal{U}$. The goal in this setting is to design an algorithm $\pi$ that minimizes the suboptimality gap for an incoming user $u_{\text{test}}$ to be served, defined as (Jin et al., 2021; Li et al., 2022):

$$\text{SubOpt}(u_{\text{test}}, \pi, \mathcal{A}_{\text{test}}) := \langle \boldsymbol{\theta}_{u_{\text{test}}}, \boldsymbol{a}_{\text{test}}^* \rangle - \langle \boldsymbol{\theta}_{u_{\text{test}}}, \boldsymbol{a}_{\pi(u_{\text{test}})} \rangle,$$

where $\boldsymbol{a}_{\text{test}}^*$ and $\boldsymbol{a}_{\pi(u_{\text{test}})}$ denote the optimal action in $\mathcal{A}_{\text{test}}$ and the action selected by $\pi$, respectively.

# 3 ALGORITHM FOR GENERAL CASES: OFF-C²LUB

This section introduces our first algorithm: Offline Connection-based Clustering of Bandits Algorithm (Off-C²LUB) and the theoretical results on its suboptimality.

## 3.1 ALGORITHM 1: OFF-C²LUB

Compared to the online clustering of bandits problem, the core challenge in Off-ClusBand lies in data insufficiency. In the online setting, algorithms start with a complete graph over all users, where each node represents a user and each edge encodes a tentative assumption that the connected users are thought to be in the same cluster (Gentile et al., 2014; Li & Zhang, 2018) and iteratively delete edges to separate users into clusters (i.e., remaining connected subgraphs). With abundant online data, this process accurately removes incorrect connecting edges. However, in the offline setting, limited data makes it difficult to precisely remove edges, often leading to the inclusion of heterogeneous users within the same cluster, which introduces bias and reduces decision quality. Therefore, building on the challenges described above, the core idea behind Off-C²LUB is to prevent

users with significantly different preference vectors from being clustered. Specifically, the algorithm treats users with few samples independently, while leveraging the clustering property only for those with a sufficient dataset $\mathcal{D}_u$. The workflow of the proposed Off-C$^2$LUB successfully avoids clustering largely heterogeneous users by ensuring that predictions are made solely using data from users who share the same (or similar) preference vector as $u_{\text{test}}$. Algorithm 1 shows the pseudo-code for Off-C$^2$LUB which consists a Cluster Phase and a Decision Phase.

---

**Algorithm 1** Off-C$^2$LUB

---

1: **Input:** User $u_{\text{test}}$, action set $\mathcal{A}_{\text{test}}$, dataset $\mathcal{D}$, parameters $\alpha > 1$, $\lambda > 0$, $\delta > 0$ and $\hat{\gamma} \geq 0$

2: **Initialization:** Construct a **null** graph $\mathcal{G} = (\mathcal{V}, \emptyset)$ where $\mathcal{V} = \mathcal{U}$, set $N_{\min} = \frac{16}{\lambda_a^2} \log\left(\frac{8Ud}{\lambda_a^2 \delta}\right)$,

   and compute $M_u$, $\boldsymbol{b}_u$, $\text{CI}_u$ as in Equation (1) and $\hat{\boldsymbol{\theta}}_u = M_u^{-1} \boldsymbol{b}_u$ for each user $u \in \mathcal{U}$

3: \\Cluster Phase

4: **for** any two users $u_1$, $u_2 \in \mathcal{V}$ **do**

5:    Connect $(u_1, u_2)$ if conditions in Equation (2) are satisfied

6: **end for**

7: Let $\mathcal{G}_{\hat{\gamma}} = (\mathcal{V}, \mathcal{E}_{\hat{\gamma}})$ denote the graph afterwards

8: **for** each user $u \in \mathcal{V}$ **do**

9:    Aggregate data and compute statistics:

$$\tilde{\mathcal{D}}_u = \bigcup_{(u,v)\in\mathcal{E}_{\hat{\gamma}}} \left\{ (\boldsymbol{a}_v^i, r_v^i) \in \mathcal{D}_v \right\} \cup \mathcal{D}_u, \ \tilde{n}_u = 1 + \sum_{v\in\mathcal{V}} \mathbb{I}\left[(u,v) \in \mathcal{E}_{\hat{\gamma}}\right]$$

$$\tilde{M}_u = \lambda \tilde{n}_u \cdot I + \sum_{(\boldsymbol{a},r)\in\tilde{\mathcal{D}}_u} \boldsymbol{a}\boldsymbol{a}^\top, \ \tilde{N}_u = \left|\tilde{\mathcal{D}}_u\right|, \ \tilde{\boldsymbol{b}}_u = \sum_{(\boldsymbol{a},r)\in\tilde{\mathcal{D}}_u} r\boldsymbol{a}, \ \tilde{\boldsymbol{\theta}}_u = \left(\tilde{M}_u\right)^{-1} \tilde{\boldsymbol{b}}_u$$

10: **end for**

11: \\Decision Phase

12: Select $\boldsymbol{a}_{\text{test}} = \underset{\boldsymbol{a}\in\mathcal{A}_{\text{test}}}{\arg\max}\left(\tilde{\boldsymbol{\theta}}_{u_{\text{test}}}^\top \boldsymbol{a} - \beta \|\boldsymbol{a}\|_{\tilde{M}_{u_{\text{test}}}^{-1}}\right), \beta = \sqrt{d\log\left(1 + \frac{\tilde{N}_{u_{\text{test}}}}{\lambda\tilde{n}_{u_{\text{test}}}\cdot d}\right) + 2\log\left(\frac{2U}{\delta}\right)} + \sqrt{\lambda}$

---

**Input and Initialization.** Algorithm 1 takes as input the test user $u_{\text{test}} \in \mathcal{U}$, the action set $\mathcal{A}_{\text{test}}$, the dataset $\mathcal{D}$ and several parameters (Line 1), which will be discussed later. The algorithm initializes the following (Line 2) for each user $u$:

$$M_u = \lambda I + \sum_{i=1}^{N_u} \boldsymbol{a}_u^i \left(\boldsymbol{a}_u^i\right)^\top, \ \boldsymbol{b}_u = \sum_{i=1}^{N_u} r_u^i \boldsymbol{a}_u^i, \ \text{CI}_u = \frac{\sqrt{d\log\left(1 + \frac{N_u}{\lambda d}\right) + 2\log\left(\frac{2U}{\delta}\right)} + \sqrt{\lambda}}{\sqrt{\tilde{\lambda}_a N_u/2}}. \quad (1)$$

Here, $M_u$ denotes the Gramian matrix regularized by $\lambda$, $\boldsymbol{b}_u$ is the moment matrix of the response variable, $\hat{\boldsymbol{\theta}}_u$ is the ridge regression estimate of $\boldsymbol{\theta}_u$ for user $u$, and $\text{CI}_u$ represents the confidence interval bound. The algorithm also defines $N_{\min}$ as the minimum number of samples required for each user during clustering. These quantities play a central role in both phases of the algorithm.

**Cluster Phase.** Instead of starting with a complete graph as in the online setting, Algorithm 1 initializes a *null* graph $\mathcal{G}$ with only vertices but no edges. During the Cluster Phase, the algorithm iterates over all user pairs $u_1$ and $u_2$ in $\mathcal{V}$, connecting them if the following condition holds:

$$\left\|\hat{\boldsymbol{\theta}}_{u_1} - \hat{\boldsymbol{\theta}}_{u_2}\right\|_2 < \hat{\gamma} - \alpha\left(\text{CI}_{u_1} + \text{CI}_{u_2}\right) \text{ and } \min\left\{N_{u_1}, N_{u_2}\right\} \geq N_{\min}. \quad (2)$$

The condition in Equation (2) ensures that the difference between the true preference vectors $\boldsymbol{\theta}_{u_1}$ and $\boldsymbol{\theta}_{u_2}$ is no greater than $\hat{\gamma}$ (see Lemma 3.2 for a detailed explanation). By only connecting users with similar estimated preference vectors, the algorithm prevents the introduction of bias that arises when users with highly divergent preference vectors are connected. Therefore, the choice of $\hat{\gamma}$ significantly affects the algorithm's performance. As discussed in Section 3.2, a smaller $\hat{\gamma}$ helps to reduce bias from heterogeneous users of $u_{\text{test}}$, but may also limit access to samples from homogeneous users. In contrast, a larger $\hat{\gamma}$ includes more samples to reduce noise, but risks introducing bias from heterogeneous users. We further discuss strategies for selecting $\hat{\gamma}$ in Section 3.3. Once all users in $\mathcal{V}$ have been checked, the graph is updated to $\mathcal{G}_{\hat{\gamma}}$, reflecting these new connections.

Next, for each user $u$, the algorithm aggregates data from the user and its **neighbors** in $\mathcal{G}_{\hat{\gamma}}$ to form a new dataset $\tilde{\mathcal{D}}_u$ (Line 9), which represents the samples from users who are likely to have the same

preference vector as the test user. Unlike traditional online algorithms that aggregate data from the entire connected component containing $u$, our approach aggregates only from one-hop neighbors. We retain the term "clustering" in our algorithm's name to stay consistent with the problem setting and prior works. This conservative strategy reduces uncontrolled bias from highly heterogeneous users (Wang et al., 2023a; Dai et al., 2024a) and is better suited to offline scenarios where data is limited, as discussed in detail in Appendix G. The algorithm then computes the necessary statistics for the decision phase. These statistics, similar to those defined in Equation (1), encompass a broader set of users with similar preferences as $u_{\text{test}}$, enhancing the algorithm's decision-making capability.

**Decision Phase.** Finally, the algorithm selects $\boldsymbol{a}_{\text{test}}$ for the test user $u_{\text{test}}$ based on the statistics $\tilde{M}_{u_{\text{test}}}$ and $\tilde{\boldsymbol{\theta}}_{u_{\text{test}}}$ (Line 9), following a pessimistic estimate of the preference vector $\boldsymbol{\theta}_{u_{\text{test}}}$ (Jin et al., 2021; Rashidinejad et al., 2021; Li et al., 2022), ensuring robust decision-making with offline data.

## 3.2 UPPER BOUND FOR OFF-C$^2$LUB

We analyze the upper bound on the suboptimality of Algorithm 1, with detailed proofs provided in Appendix D.1 and D.2. Since the algorithm aggregates user data based on the graph $\mathcal{G}_{\hat{\gamma}}$ constructed in Line 7, we define $\mathcal{V}_{\hat{\gamma}}(u), \mathcal{R}_{\hat{\gamma}}(u)$ and $\mathcal{W}_{\hat{\gamma}}(u)$ as Table 2.

Table 2: Summary of neighbor set notations.

| Notation | Definition | Interpretation |
|---|---|---|
| $\mathcal{V}_{\hat{\gamma}}(u)$ | $\{u\} \cup \{v \mid (u,v) \in \mathcal{E}_{\hat{\gamma}}\}$ | Set of user $u$ together with all its neighbors in graph $\mathcal{G}_{\hat{\gamma}}$. |
| $\mathcal{R}_{\hat{\gamma}}(u)$ | $\{v \mid v \in \mathcal{V}_{\hat{\gamma}}(u), \boldsymbol{\theta}_u = \boldsymbol{\theta}_v\}$ | Set of $u$ and its *homogeneous neighbors*, i.e., users in $\mathcal{V}_{\hat{\gamma}}(u)$ with same preference vectors. Their data can be safely aggregated with $u$'s without introducing bias. |
| $\mathcal{W}_{\hat{\gamma}}(u)$ | $\{v \mid v \in \mathcal{V}_{\hat{\gamma}}(u), \boldsymbol{\theta}_u \neq \boldsymbol{\theta}_v\}$ | Set of $u$'s *heterogeneous neighbors*, i.e., users in $\mathcal{V}_{\hat{\gamma}}(u)$ with different preference vectors. Aggregating their data with $u$'s may introduce bias and thus should be controlled. |

Thus $\mathcal{V}_{\hat{\gamma}}(u) = \mathcal{R}_{\hat{\gamma}}(u) \cup \mathcal{W}_{\hat{\gamma}}(u)$ and $\mathcal{R}_{\hat{\gamma}}(u) \cap \mathcal{W}_{\hat{\gamma}}(u) = \emptyset$ hold. To formalize cluster separation, we introduce the *heterogeneity gap* (Gentile et al., 2014; Li et al., 2019; 2025; Wang et al., 2025) as:

**Definition 3.1** (Heterogeneity Gap). The preference vectors of users from different clusters are separated by at least $\gamma$; that is, for any $u$ and $v$ in different clusters, $\|\boldsymbol{\theta}_u - \boldsymbol{\theta}_v\|_2 \geq \gamma > 0$.

In Algorithm 1, the choice of $\hat{\gamma}$ influences both $\mathcal{R}_{\hat{\gamma}}(u)$ and $\mathcal{W}_{\hat{\gamma}}(u)$, and thus directly impacts both performance and theoretical guarantees. The following lemma formalizes these effects, while Theorem 3.4 establishes the resulting suboptimality bound.

**Lemma 3.2** (Estimations of sets $\mathcal{R}_{\hat{\gamma}}(u)$ and $\mathcal{W}_{\hat{\gamma}}(u)$). *For inputs $\alpha \geq 1, \lambda > 0,$ and $\delta \in (0,1)$ satisfying $\lambda \leq d \log \left(1 + \frac{\min_u\{N_u\}}{\lambda d}\right) + 2\log\left(\frac{2U}{\delta}\right)$ and $\delta \leq \frac{2U}{1+\max_u\{N_u\}/(\lambda d)}$, there exist some $\alpha_r \in \left[\frac{\sqrt{\lambda_a}}{4(\alpha+1)\sqrt{\log(2U/\delta)}\max\{2,d\}}, \frac{\sqrt{\lambda_a}}{2\alpha\sqrt{\log(2U/\delta)}}\right)$ and $\alpha_w \in \left(0, \frac{\sqrt{\lambda_a}}{2(\alpha-1)\sqrt{\log(2U/\delta)}}\right]$ such that sets $\mathcal{R}_{\hat{\gamma}}(u)$ and $\mathcal{W}_{\hat{\gamma}}(u)$ defined in Table 2 can be represented as:*

$$\text{If } N_u \geq N_{\min} : \mathcal{R}_{\hat{\gamma}}(u) = \{u\} \cup \left\{v \,\Big|\, \boldsymbol{\theta}_u = \boldsymbol{\theta}_v, \; \frac{1}{\sqrt{N_u}} + \frac{1}{\sqrt{N_v}} \leq \alpha_r \hat{\gamma}, \; N_v \geq N_{\min}\right\},$$

$$\mathcal{W}_{\hat{\gamma}}(u) = \left\{v \,\Big|\, \gamma \leq \|\boldsymbol{\theta}_u - \boldsymbol{\theta}_v\|_2 \leq \hat{\gamma}, \; \frac{1}{\sqrt{N_u}} + \frac{1}{\sqrt{N_v}} \leq \alpha_w \varepsilon, \; N_v \geq N_{\min}\right\};$$

$$\text{Otherwise} : \mathcal{R}_{\hat{\gamma}}(u) = \{u\}, \mathcal{W}_{\hat{\gamma}}(u) = \emptyset,$$

*with probability at least $1 - \delta$, where $N_{\min}$ is as defined in Line 2 of Algorithm 1 and $\varepsilon = \hat{\gamma} - \gamma$.*

*Remark* 3.3 (Interpretation of Lemma 3.2). As defined in Table 2, set $\mathcal{R}_{\hat{\gamma}}(u)$ contains user $u$ and its homogeneous neighbors while $\mathcal{W}_{\hat{\gamma}}(u)$ represents $u$'s heterogeneous neighbors. In Lemma 3.2, the distinction between the two cases (whether $N_u \geq N_{\min}$) follows directly from the cluster condition in Equation (2). Specifically, when $u$ does not meet the minimum data requirement $N_{\min}$, no connections are formed, and the algorithm defaults to using only $u$'s own data. This threshold ensures sufficient coverage of $\mathbb{R}^d$ under Assumption 2.1, providing essential information along each dimension for preference estimation. When $N_u \geq N_{\min}$, the constraints on $1/\sqrt{N_u} + 1/\sqrt{N_v}$ for both $\mathcal{R}_{\hat{\gamma}}(u)$ and $\mathcal{W}_{\hat{\gamma}}(u)$ reflect the principle that only users with sufficient data and reliable estimates

should contribute to decision-making. Increasing $\hat{\gamma}$ (and thus $\varepsilon$) relaxes these constraints, potentially incorporating more homogeneous samples but at the risk of introducing larger bias. Moreover, the constraint $\gamma \leq \|\boldsymbol{\theta}_u - \boldsymbol{\theta}_v\|_2 \leq \hat{\gamma}$ ensures that $\mathcal{W}_{\hat{\gamma}}(u)$ only includes users with relatively similar preferences where $\hat{\gamma}$ is a tunable input. A larger $\hat{\gamma}$ increases such bias and enlarges $\mathcal{W}_{\hat{\gamma}}(u)$.

**Theorem 3.4.** *With the same conditions as Lemma 3.2, the suboptimality for Off-$C^2$LUB satisfies:*

$$\text{SubOpt}(u_{\text{test}}, \text{Off-C}^2\text{LUB}, \mathcal{A}_{\text{test}}) \leq \tilde{O}\left(\sqrt{\frac{d}{\tilde{\lambda}_a N_{\mathcal{V}_{\hat{\gamma}}(u_{\text{test}})}}} + \frac{\eta_{\mathcal{W}_{\hat{\gamma}}(u_{\text{test}})}\hat{\gamma}}{\tilde{\lambda}_a}\right), \tag{3}$$

*with probability at least $1 - 2\delta$, where $\tilde{O}$ omits logarithmic terms and $\eta_{\mathcal{W}_{\hat{\gamma}}(u)} = \frac{|\mathcal{W}_{\hat{\gamma}}(u)|\lambda + N_{\mathcal{W}_{\hat{\gamma}}(u)}}{|\mathcal{V}_{\hat{\gamma}}(u)|\lambda + N_{\mathcal{V}_{\hat{\gamma}}(u)}}$ denotes the fraction of heterogeneous samples over all utilized samples for $u$.*

*Remark* 3.5 (Analysis of Theorem 3.4). The terms in Theorem 3.4 reflect both **noise** and **bias**. The first term, caused by the limited number of aggregated samples, represents the **noise** from all neighbors of $u_{\text{test}}$ in $\mathcal{V}_{\hat{\gamma}}(u_{\text{test}})$. This noise decreases as the total number of samples in $\mathcal{V}_{\hat{\gamma}}(u_{\text{test}})$ grows, which can be achieved selecting a larger $\hat{\gamma}$. The second term reflects the **bias** from including samples of heterogeneous neighbors, which scales with both $\hat{\gamma}$ and the fraction of heterogeneous samples $\eta_{\mathcal{W}_{\hat{\gamma}}(u_{\text{test}})}$. As noted in Remark 3.3, enlarging $\hat{\gamma}$ expands $\mathcal{W}_{\hat{\gamma}}(u_{\text{test}})$ and thus potentially increases $\eta_{\mathcal{W}_{\hat{\gamma}}(u_{\text{test}})}$. Therefore, controlling $\hat{\gamma}$ is key to balancing two competing effects: expanding $\mathcal{V}_{\hat{\gamma}}(u_{\text{test}})$ to reduce noise, and restricting $\mathcal{W}_{\hat{\gamma}}(u_{\text{test}})$ to limit bias. The dependence on $\tilde{\lambda}_a$ highlights the link between suboptimality and action regularity: a larger $\tilde{\lambda}_a$ implies more information per sample and hence smaller suboptimality. Importantly, the form of Theorem 3.4 reflects the fundamental limited-data challenge of the offline setting: unlike online scenarios, where additional samples can be collected to reduce noise and accurately estimate user preferences to avoid bias, the offline setting enforces a crucial tradeoff between noise and bias, making the choice of $\hat{\gamma}$ particularly critical.

*Remark* 3.6 (Discussions on $\alpha_r$ and $\alpha_w$). The parameters $\alpha_r$ and $\alpha_w$ in Lemma 3.2 are not fixed constants but are bounded within setting-dependent ranges. This design reflects the practical difficulty of determining each user's exact neighbors during our theoretical analysis due to the uncertainty and noise in offline datasets, while still ensuring that the cardinalities of $\mathcal{R}_{\hat{\gamma}}(u)$ and $\mathcal{W}_{\hat{\gamma}}(u)$ can be estimated reliably. These parameters therefore play a critical role in ensuring the generality of our results. Intuitively, by choosing $\alpha_r$ at the lower end of its range to conservatively approximate a subset of homogeneous neighbors and $\alpha_w$ at the upper end of its range to conservatively approximate a superset of heterogeneous neighbors, we can establish a concrete bound that remains parameter-fixed under uncertainty. We defer the refinement of these parameters, along with the derivation of a fixed parameter form of Theorem 3.4, to Appendix C.1 due to space constraints.

## 3.3 Strategies for Choosing $\hat{\gamma}$

As demonstrated in Theorem 3.4, the performance of Algorithm 1 critically depends on the choice of $\hat{\gamma}$ where a carefully chosen $\hat{\gamma}$ balances noise and bias. Below we outline strategies for selecting $\hat{\gamma}$ under different scenarios. Figure 1 shows the influence of different choices for $\hat{\gamma}$.

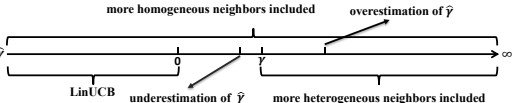

Figure 1: Influence of Different $\hat{\gamma}$.

**Case 1: Exact Information of $\gamma$.** We set $\hat{\gamma} = \gamma$ when the exact value of $\gamma$ is known. This ensures $\varepsilon = 0$ in Theorem 3.4, so that only users from the same true cluster as $u_{\text{test}}$ are included, i.e., $\eta_{\mathcal{W}_\gamma(u_{\text{test}})} = 0$. The resulting suboptimality is formalized in Corollary 3.7.

**Corollary 3.7** (Theorem 3.4 for Accurate $\gamma$). *With the same notations as Theorem 3.4, if $\hat{\gamma} = \gamma$ in Algorithm 1, the suboptimality gap satisfies:*

$$\text{SubOpt}(u_{\text{test}}, \text{Off-C}^2\text{LUB}, \mathcal{A}_{\text{test}}) \leq \tilde{O}\left(\sqrt{\frac{d}{\tilde{\lambda}_a N_{\mathcal{V}_\gamma(u_{\text{test}})}}}\right).$$

*Remark* 3.8 (Analysis of Corollary 3.7). Corollary 3.7 implies that Algorithm 1 achieves suboptimality only with noise. This is because no users from outside $u_{\text{test}}$'s true cluster are included, eliminating bias entirely. Furthermore, Algorithm 1 only connects homogeneous users based on similar preference vectors, avoiding the biased inclusion of heterogeneous users. Note that choosing $\hat{\gamma} < \gamma$ can also yield a bias-free suboptimality as in Corollary 3.7.

**Case 2: No Prior Information on $\gamma$.** The more general and challenging case arises when no prior knowledge of $\gamma$ is available, requiring $\hat{\gamma}$ to be estimated directly from data. To guide this estimation, we define the following statistics:

$$\Gamma(u,v) = \left\|\hat{\boldsymbol{\theta}}_u - \hat{\boldsymbol{\theta}}_v\right\|_2 - \alpha(\text{CI}_u + \text{CI}_v), \; \tilde{\Gamma}(u,v) = \left\|\hat{\boldsymbol{\theta}}_u - \hat{\boldsymbol{\theta}}_v\right\|_2 + \alpha(\text{CI}_u + \text{CI}_v), \quad (4)$$

$$M(u) = \{v \in \mathcal{V} \setminus \{u\} \mid \Gamma(u,v) > 0\},$$

where $\text{CI}_u$ is as defined in Equation (1). Since $\Gamma(u,v) \leq \|\boldsymbol{\theta}_u - \boldsymbol{\theta}_v\|_2$ for $\alpha \geq 1$, it serves as a lower bound on the true preference gap. If $\Gamma(u,v) \leq 0$, $u$ and $v$ are either in the same cluster or lack sufficient data to be distinguished. Hence $M(u)$ contains users from different clusters, while those outside are either homogeneous or uncertain. When $M(u_{\text{test}}) = \emptyset$, it is natural to set $\hat{\gamma} = 0$, reducing Algorithm 1 to LinUCB (Abbasi-Yadkori et al., 2011) based solely on $\mathcal{D}_{u_{\text{test}}}$.

Below, we propose two policies for choosing $\hat{\gamma}$: the *underestimation policy* selects a smaller value to limit bias but at the cost of more noise, while the *overestimation policy* makes the opposite trade-off:

$$\hat{\gamma} = \begin{cases} \mathbb{I}\{M(u_{\text{test}}) \neq \emptyset\} \cdot \min_{v \in M(u_{\text{test}})} \Gamma(u_{\text{test}}, v), & \textbf{(Underestimation Policy)}, \\ \mathbb{I}\{M(u_{\text{test}}) \neq \emptyset\} \cdot \min_{v \in M(u_{\text{test}})} \tilde{\Gamma}(u_{\text{test}}, v), & \textbf{(Overestimation Policy)}, \end{cases} \quad (5)$$

*Remark* 3.9 (Suitable Scenarios for Both Policies). The key difference between the two policies lies in whether we underestimate ($\Gamma(u,v)$) or overestimate ($\tilde{\Gamma}(u,v)$) the preference gap between two users. Intuitively, the underestimation policy selects a relatively small $\hat{\gamma}$, which reduces bias but increases noise according to Lemma 3.2, while the overestimation policy makes the opposite tradeoff. Therefore, the underestimation policy is more suitable when avoiding bias is more critical, making it particularly valuable in safety-critical applications such as medical treatment or financial decision-making, where even small biases could lead to harmful outcomes. By contrast, the overestimation policy is preferable when aggregating large, diverse data is essential. For example, in recommender systems with high-dimensional user preferences, pooling additional data helps stabilize estimates and improves accuracy, even at the cost of introducing moderate bias. Detailed theoretical analysis and further discussion of both policies are deferred to Appendix C.2 due to space constraints.

## 4 ALGORITHM FOR SUFFICIENT DATASET: OFF-CLUB

When $\gamma$ is unknown, Off-C$^2$LUB faces a trade-off: a quite small $\hat{\gamma}$ increases noise, while a large one introduces bias, as shown in Section 3.3. This motivates us to propose Offline Clustering of Bandits (Off-CLUB) that avoids estimating $\gamma$ and achieves strong performance under sufficient data.

### 4.1 ALGORITHM 2: OFF-CLUB

Off-CLUB shares the same two-phase structure as Off-C$^2$LUB, consisting of a Cluster Phase and a Decision Phase. Due to space constraints, full details and pseudo-code (Algorithm 2) are deferred to Appendix G. The main difference lies in graph construction: Off-CLUB starts with a complete graph and removes edges between users with dissimilar preferences, whereas Off-C$^2$LUB begins with an empty graph and adds edges only when sufficient similarity is detected.

The algorithm takes similar inputs to Algorithm 1 but does not require $\hat{\gamma}$. During the Cluster Phase, it iterates through all edges $(u_1, u_2) \in \mathcal{E}$ and removes those that satisfying:

$$\left\|\hat{\boldsymbol{\theta}}_{u_1} - \hat{\boldsymbol{\theta}}_{u_2}\right\|_2 > \alpha\left(\text{CI}_{u_1} + \text{CI}_{u_2}\right), \quad (6)$$

indicating that users $u_1$ and $u_2$ belong to different clusters with high probability. After edge removal, the resulting graph $\tilde{\mathcal{G}}$ correctly clusters users with sharing preference vectors, provided that all users have sufficient samples as discussed in Section 4.2. Finally, the subsequent steps, including data aggregation, computation of statistics, and the Decision Phase, are the same to those in Off-C$^2$LUB.

### 4.2 UPPER BOUND FOR OFF-CLUB UNDER SUFFICIENT DATA

The effectiveness of Off-CLUB relies on accurately removing edges between users in different clusters, requiring $\text{CI}_u$ to be sufficiently small. This is typically satisfied when users have many samples, while users with few samples may remain incorrectly connected, grouping heterogeneous users together. We therefore formalize the data sufficiency condition for the success of Off-CLUB:

**Assumption 4.1** ($\delta$-Sufficient Dataset). The dataset $\mathcal{D}_u$ is called $\delta$-sufficient for user $u$ if $N_u \geq \max\left\{\frac{16}{\tilde{\lambda}_a^2}\log\left(\frac{8dU}{\tilde{\lambda}_a^2\delta}\right), \frac{512d}{\gamma^2\tilde{\lambda}_a}\log\left(\frac{2U}{\delta}\right)\right\}$ for some $\delta \in (0,1)$. Furthermore, we assume that the full dataset $\mathcal{D} = \{\mathcal{D}_u\}_{u=1}^U$ is **completely** $\delta$-**sufficient**, i.e., each user's dataset $\mathcal{D}_u$ is $\delta$-sufficient.

Assumption 4.1 guarantees data sufficiency to estimate preferences and cluster accurately. Theorem 4.2 (proof in Appendix D.5) guarantees the performance for Off-CLUB under this assumption.

**Theorem 4.2** (Upper Bound for Off-CLUB). *For any $\delta \in (0, \frac{1}{2})$, assume $\mathcal{D}$ is completely $\delta$-sufficient. Then for $\alpha = 1$, $\lambda > 0$, with probability at least $1 - 2\delta$, the suboptimality gap satisfies*

$$\text{SubOpt}(u_{\text{test}}, \text{Off-CLUB}, \mathcal{A}_{\text{test}}) \leq \tilde{O}\left(\sqrt{\frac{d}{\tilde{\lambda}_a N_{\mathcal{V}(j_{u_{\text{test}}})}}}\right).$$

*Remark* 4.3 (Analysis of Theorem 4.2). A key requirement for achieving this bound is the sufficiency of data for each user (Assumption 4.1). When this assumption is violated, Off-CLUB may incorporate samples from **highly biased users** with preference vectors largely different with $u_{\text{test}}$, degrading performance. In contrast, as shown in Theorem 3.4, Off-C$^2$LUB mitigates this issue by restricting heterogeneous neighbors to a much smaller set $\mathcal{W}_{\hat{\gamma}}(u_{\text{test}})$, controlled by $\hat{\gamma}$. Thus, while Off-CLUB performs well with sufficient data, Off-C$^2$LUB remains a more robust algorithm in general scenarios where Assumption 4.1 does not hold. We validate this with experiments in Section 5.

### 4.3 LOWER BOUND FOR OFF-CLUSBAND PROBLEM AND OPTIMALITY

To facilitate further comparison between Off-CLUB's and Off-C$^2$LUB's performance, we derive a theoretical lower bound for the Off-ClusBand problem. The proof is in Appendix D.6.

**Theorem 4.4** (Off-ClusBand Lower Bound). *Given any test user $u_{\text{test}} \in \mathcal{U}$, consider the set of Off-ClusBand instances: $\mathcal{I}_{u_{\text{test}}} = \left\{(\mathcal{D}, \Theta, R) \,\middle|\, \left\|\left(M^{\mathcal{V}(j_{u_{\text{test}}})}\right)^{-\frac{1}{2}} \boldsymbol{a}_{\text{test}}^*\right\|_2 \leq 2\sqrt{2},\ R \text{ is 1-subgaussian}\right\}$, where $\Theta = \{\boldsymbol{\theta}_u\}_{u \in \mathcal{V}}$, $\boldsymbol{a}_{\text{test}}^*$ is the optimal action for user $u_{\text{test}}$, and $M^{\mathcal{V}(j_{u_{\text{test}}})} = \sum_{v \in \mathcal{V}_{j_{u_{\text{test}}}}} M_v$. If the dataset satisfies $N_{u_{\text{test}}} \geq 8d$, the following lower bound holds:*

$$\inf_\pi \sup_{\mathcal{Q} \in \mathcal{I}_{u_{\text{test}}}} \mathbb{E}[\text{SubOpt}_{\mathcal{Q}}(u_{\text{test}}, \pi, \mathcal{A}_{\text{test}})] \geq \sqrt{8d/N_{\mathcal{V}(j_{u_{\text{test}}})}}.$$

*Remark* 4.5 (Comparison Between Upper and Lower Bounds). The lower bound in Theorem 4.4 shows that the optimal strategy for Off-ClusBand is to use only samples from users in the same cluster as $u_{\text{test}}$ while excluding heterogeneous users. Recall that under Assumption 4.1, Theorem 4.2 gives a bound scaling with $1/\sqrt{N_{\mathcal{V}(j_{u_{\text{test}}})}}$. Under the same assumption, Algorithm 1 (Off-C$^2$LUB) with $\hat{\gamma} = \gamma$ achieves the same result as Theorem 4.2 by **only** connecting all homogeneous users (as shown in Lemma 3.2 with $\alpha = 1$). This matches the lower bound up to logarithmic factors in terms of the total number of utilized samples $N_{\mathcal{V}(j_{u_{\text{test}}})}$, demonstrating near-optimal performance of both algorithms under Assumption 4.1. When data is sparse, however, Off-CLUB often clusters highly heterogeneous users and introduces bias, whereas Off-C$^2$LUB reduces this risk by restricting heterogeneous neighbors to $\mathcal{W}_{\hat{\gamma}}(u_{\text{test}})$, explaining the strength of Off-C$^2$LUB with insufficient data.

## 5 EXPERIMENTS

**Experimental Setup.** We evaluate on both *synthetic* and *real-world datasets*, including Yelp (yel, 2023) and MovieLens (Harper & Konstan, 2015). We fix $U = 1\text{k}$ users with preference dimension $d = 20$, and consider dataset sizes from 5k–100k (insufficient data) and 0.2M–1M (sufficient data), splitting each dataset evenly for training and evaluation. Synthetic users are divided into $J = 10$ clusters under two sampling schemes: *Equal Distribution* (uniform over users) and *Semi-Random Distribution* (uniform within clusters, non-uniform across clusters). For real-world data, we keep the top 1,000 users and items by rating count and derive preference vectors via SVD (Li et al., 2019). All results are averaged over 10 random runs. Baselines include bandit methods (LinUCB (Abbasi-Yadkori et al., 2011), CLUB (Gentile et al., 2014), SCLUB (Li et al., 2019)) and clustering methods (DBSCAN (Schubert et al., 2017), XMeans (Pelleg et al., 2000), ARMUL (Duan & Wang, 2023)), adapted to the offline setting. Offline data is generated either by random selection (Figure 2) or LinUCB-based item selection (Appendix F.3). Further details are given in Appendices F.1–F.3.

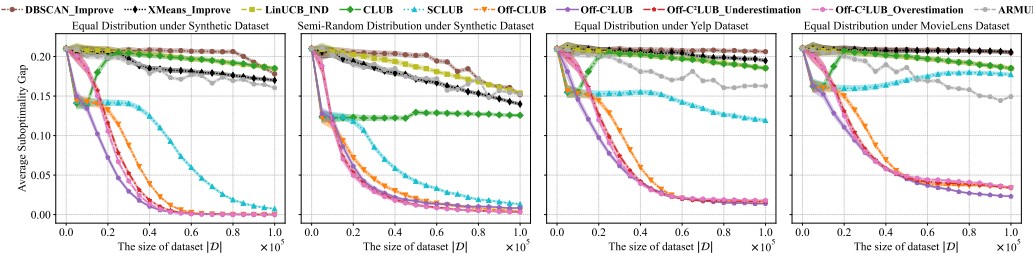

Figure 2: Comparisons of our algorithms with baselines under different user distributions and datasets. Off-C²LUB and its variants consistently outperform Off-CLUB and other baselines.

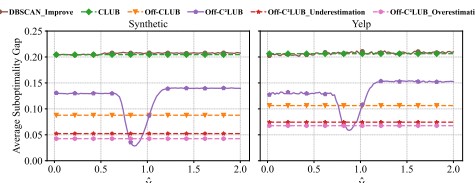

Figure 3: Suboptimality under varying $\hat{\gamma}$, showing Off-C²LUB is near-optimal with our overestimation or underestimation strategies.

Table 3: Comparisons under sufficient data show Off-CLUB and Off-C²LUB reach near-zero suboptimality, outperforming baselines.

| Algorithm \ Dataset size | 0.2M | 0.4M | 0.6M | 0.8M | 1M |
|---|---|---|---|---|---|
| DBSCAN_Improve | 0.130230 | 0.058202 | 0.024840 | 0.011893 | 0.006512 |
| XMeans_Improve | 0.130088 | 0.058248 | 0.024820 | 0.011908 | 0.007534 |
| CLUB | 0.144795 | 0.066638 | 0.028110 | 0.013647 | 0.007557 |
| Off-CLUB | **0.000037** | **0.000018** | 0.000013 | **0.000009** | **0.000007** |
| LinUCB_IND | 0.144894 | 0.066417 | 0.028238 | 0.013664 | 0.007580 |
| SCLUB | 0.001633 | 0.000447 | 0.000165 | 0.000095 | 0.000057 |
| ARMUL | 0.146900 | 0.109480 | 0.078815 | 0.055882 | 0.051504 |
| Off-C²LUB | 0.000070 | 0.000020 | **0.000012** | **0.000009** | **0.000007** |
| Off-C²LUB_Underestimation | **0.000037** | 0.000019 | 0.000013 | 0.000010 | 0.000008 |
| Off-C²LUB_Overestimation | 0.000042 | **0.000018** | **0.000012** | **0.000009** | **0.000007** |

**Performance with Insufficient Dataset.** To illustrate the performance under insufficient data, we report the average suboptimality gap across user distributions and dataset sizes (Figure 2). For Off-C²LUB, $\hat{\gamma}$ is optimized under the equal distribution scenario and applied to the semi-random distribution. With $|\mathcal{D}| \in [20k, 100k]$, both Off-C²LUB variants significantly reduce the suboptimality gap. On the synthetic dataset, Off-C²LUB_Overestimation improves over Off-CLUB by 44.2% and over other baselines by at least 77.5%, while Off-C²LUB_Underestimation achieves 34.0% and 73.3%, respectively. On Yelp, the corresponding improvements are 19.8% and 73.9% for Overestimation, and 17.9% and 73.3% for Underestimation. On MovieLens, they are 8.6% and 67.9% for Overestimation, and 9.3% and 68.0% for Underestimation. Overall, Off-C²LUB achieves the best performance, while Off-CLUB, though less effective, still outperforms classical clustering baselines.

**Performance with Sufficient Data.** Table 3 reports the performance of different algorithms under sufficient data. The bold values mark the optimal suboptimal gaps (rounded to six decimals) for each dataset size. Our proposed Off-CLUB and Off-C²LUB consistently outperform all other baselines in this regime, with the $\hat{\gamma}$ selection strategies for Off-C²LUB also guarantee near optimal performances. This experiment shows that with sufficient data, both Off-CLUB and Off-C²LUB approach to near-zero suboptimality as more offline data is provided, outperforming existing methods.

**Improvement with Estimated $\hat{\gamma}$.** We evaluate various strategies under insufficient data using synthetic and Yelp datasets (Figure 3) with $|\mathcal{D}| = 30k$. The results highlight that the suboptimality gap of Off-C²LUB increases when $\hat{\gamma}$ is set either too small or too large, which aligns with our theoretical analysis of the noise–bias tradeoff in Lemma 3.2, Theorem 4.2, and Figure 1. On both the synthetic and Yelp datasets, the proposed overestimation and underestimation strategies achieve near-optimal performance. Additional experiments are provided in Appendix F, where we further examine the robustness of our methods to varying dataset sizes and datasets.

## 6 CONCLUSION

In this paper, we address the offline clustering of bandits (Off-ClusBand) problem, proposing two algorithms: Off-C²LUB, effective in general scenarios, and Off-CLUB, which excels with sufficient data. We analyze the influence of $\hat{\gamma}$ on Off-C²LUB, offering policies for cases where $\gamma$ is known or unknown. Additionally, we establish a theoretical lower bound for Off-ClusBand. Experiments on synthetic and real-world datasets validate the robustness and efficiency of our methods. Future work may explore more general reward structures beyond linear models that this paper does not cover, such as nonlinear or nonparametric formulations. It is also promising to consider hybrid settings that combine offline data with online interaction for improved adaptability in real-world scenarios.

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

## A RELATED WORKS

**Offline Bandits.** Offline multi-armed bandits (MAB), introduced by Shivaswamy & Joachims (2012), are a specialized variant of offline reinforcement learning (RL) that utilize pre-collected data for decision-making based on confidence balls for regression problems (Javanmard & Montanari, 2014; Zhang & Zhang, 2014; Cai & Guo, 2017; Duan & Wang, 2023) where Duan & Wang (2023) also considers parameter estimation with heterogeneous data. These methods have been applied in domains such as robotics (Kumar et al., 2020), autonomous driving (Yurtsever et al., 2020), and education (Singla et al., 2021). Some offline MAB studies consider scenarios where the offline distribution remains consistent with the online rewards (Bu et al., 2020; Banerjee et al., 2022), while others address settings with reward distribution shifts (Zhang et al., 2019; Cheung & Lyu, 2024). Research has also explored offline data provided sequentially (Gur & Momeni, 2022) or in distinct groups (Bouneffouf et al., 2019; Ye et al., 2020; Tennenholtz et al., 2021). Among these, Li et al. (2022); Wang et al. (2024) focus on offline contextual linear bandits, which are most closely aligned with our work. Recent works have explored offline RL with homogeneous agents (Woo et al., 2023; 2024), as well as settings involving heterogeneous agents (Zhou et al., 2024), where each agent serves a heterogeneous user, or heterogeneous data resources (Wang et al.; Shi et al., 2023). Our work addresses offline bandits with multiple data resources for heterogeneous users and introduces a technique to identify and merge similar datasets for more accurate predictions, setting it apart from previous studies.

**Online Clustering of Bandits.** Online clustering of bandits have been extensively studied across various settings and applications, beginning with Gentile et al. (2014) and including studies linking clusters to items for recommendations (Li et al., 2016a;b; Gentile et al., 2017) and exploring non-linear reward functions (Nguyen et al., 2020; Ban et al., 2024). Distributed bandit clustering has been a focus in (Korda et al., 2016; Cherkaoui et al., 2023), while extensions to dependent arms were explored in (Wang et al., 2019). Several works focus on the way to simplify the basic assumptions (Li et al., 2019; 2025). Applications range from recommendation systems (Yang & Toni, 2018; Yan et al., 2022) to vehicular edge computing and dynamic pricing (Lin et al., 2022; Miao et al., 2022). Recent studies have also addressed privacy concerns (Liu et al., 2022), dynamic cluster switching (Nguyen & Lauw, 2014), unified clustering with non-stationary bandits (Li et al., 2021), corrupted user detection (Wang et al., 2023b) and conversational (Wu et al., 2021; Li et al., 2023; Dai et al., 2024a;b). In contrast to the online setting, where infinite samples can be drawn from unlimited online data, a key challenge in our setting lies in addressing scenarios with limited offline data. To the best of our knowledge, this is the first work to explore clustering bandits specifically in the offline bandit learning setting.

## B ADDITIONAL DISCUSSIONS ON ACTION REGULARITY ASSUMPTIONS

In this appendix, we provide a detailed discussion of Assumption 2.1. This assumption is standard in the clustering of bandits literature (Gentile et al., 2014; Li & Zhang, 2018; Li et al., 2019; Liu et al., 2022; Wang et al., 2023a; Dai et al., 2024a; Li et al., 2025; Wang et al., 2025), although the precise formulation varies some across works. The version we adopt in this paper—where each offline action is drawn from a candidate set $\mathcal{S}_u^i$ of bounded size $S$ and the algorithm is assumed to know the smoothed regularity parameter $\tilde{\lambda}_a$—follows the most recent line of research (Wang et al., 2023a; Dai et al., 2024a; Wang et al., 2025). To highlight the differences, we first introduce an alternative assumption that can remove the finite candidate set requirement or the need for $\tilde{\lambda}_a$, at the cost of stronger independence or bounded-variance conditions. We then discuss how our analysis extends to cases where no regularity assumption is imposed.

**Assumption B.1** (Action Regularity (Alternative Form)). Let $\rho$ be a distribution over $\{\boldsymbol{a} \in \mathbb{R}^d : \|\boldsymbol{a}\|_2 \leq 1\}$ such that $\mathbb{E}_{\boldsymbol{a} \sim \rho}[\boldsymbol{a}\boldsymbol{a}^\top]$ is full rank with minimum eigenvalue $\lambda_a > 0$. Each action $\boldsymbol{a}_u^i$ in dataset $\mathcal{D}_u$ is either: (i) independently and identically drawn from $\rho$, or (ii) selected from a candidate set $\mathcal{S}_u^i$, where actions in $\mathcal{S}_u^i$ are independently drawn from $\rho$ and, for any unit vector $\boldsymbol{\theta} \in \mathbb{R}^d$, the random variable $(\boldsymbol{\theta}^\top \boldsymbol{a})^2$ has sub-Gaussian tails with variance $\sigma^2 \leq \lambda_a^2/(8\log(4|\mathcal{S}_u^i|))$. We assume $\lambda_a$ is known to the algorithm.

One crucial distinction between Assumption B.1 and Assumption 2.1 lies in the prior knowledge required: the former requires $\lambda_a$, whereas the latter uses the smoothed regularity parameter $\tilde{\lambda}_a$.

Table 4: Summary of Notations for Off-ClusBand

| Notation | Description |
|---|---|
| $\mathcal{U} = \{1, \ldots, U\}$ | Set of all users; $U$ is the total number of users |
| $\boldsymbol{\theta}_u \in \mathbb{R}^d$ | Preference vector of user $u$ |
| $\mathcal{V}$ | Set of vertices in the user graph (same as $\mathcal{U}$) |
| $\mathcal{V}(j)$ | Set of users in cluster $j$; clusters are disjoint |
| $J$ | Total number of (unknown) clusters |
| $j_u$ | Cluster index to which user $u$ belongs |
| $\boldsymbol{\theta}^j$ | Shared preference vector of users in cluster $j$ |
| $\mathcal{D}_u = \{(\boldsymbol{a}_u^i, r_u^i)\}_{i=1}^{N_u}$ | Offline dataset for user $u$, containing $N_u$ samples |
| $\boldsymbol{a}_u^i \in \mathbb{R}^d$ | Action taken in the $i$-th sample of user $u$, with $\|\boldsymbol{a}_u^i\|_2 \leq 1$ |
| $r_u^i \sim R(\boldsymbol{a}_u^i)$ | Reward corresponding to action $\boldsymbol{a}_u^i$, drawn from a 1-subgaussian distribution |
| $\mathbb{E}[r_u^i] = \langle \boldsymbol{\theta}_u, \boldsymbol{a}_u^i \rangle$ | Expected reward of user $u$ on action $\boldsymbol{a}_u^i$ |
| $\rho$ | Distribution over the action set, supported on $\|\boldsymbol{a}\|_2 \leq 1$ |
| $\lambda_a$ | Minimum eigenvalue of $\mathbb{E}_{\boldsymbol{a} \sim \rho}[\boldsymbol{a}\boldsymbol{a}^\top]$ (action regularity) |
| $\mathcal{A}_{\text{test}} \subseteq \mathbb{R}^d$ | Action set available for the test-time decision |
| $u_{\text{test}}$ | Test user to be served |
| $\pi$ | Algorithm that selects actions for users |
| $\boldsymbol{a}_{\pi(u)}$ | Action selected by algorithm $\pi$ for user $u$ |
| $\boldsymbol{a}_{\text{test}}^*$ | Optimal action in $\mathcal{A}_{\text{test}}$ for $u_{\text{test}}$ |
| $\text{SubOpt}(u_{\text{test}}, \pi, \mathcal{A}_{\text{test}})$ | Suboptimality gap: difference between optimal and chosen reward for $u_{\text{test}}$ |
| $M_u$ | Gramian matrix for user $u$ regularized by the input parameter $\lambda$ |
| $\boldsymbol{b}_u$ | Moment matrix of regressand by regressors for user $u$ |
| $\text{CI}_u$ | Confidence interval for user $u$ |
| $\mathcal{R}_{\hat{\gamma}}(u)$ | User $u$ itself and its homogeneous neighbors |
| $\mathcal{W}_{\hat{\gamma}}(u)$ | User $u$'s heterogeneous neighbors |
| $\eta_{\mathcal{W}_{\hat{\gamma}}(u)}$ | Fraction of samples from heterogeneous neighbors of $u$ |

When Assumption 2.1 does not hold but Assumption B.1 does, all our theoretical claims (such as Theorem 3.4 and Theorem 4.2) remain valid after replacing every occurrence of $\tilde{\lambda}_a$ with $\lambda_a$ in the pseudo-code and analysis of Algorithms 1 and 2, according to Lemma E.2 and E.3.

**Interpretation of the Alternative Form.** Assumption B.1 accommodates two ways of generating offline data:

**(1) The i.i.d. action model.** Each action is drawn i.i.d. from $\rho$, as motivated by the online exploration phase proposed by algorithms in recent online clustering of bandits work (Li et al., 2025), and widely adopted in offline RL/bandits literature (Rashidinejad et al., 2021; Liu et al., 2025). This setting is appropriate when offline data are sampled randomly from a continuous or otherwise infinite action space. However, it does not capture cases where actions are selected by a logging policy (e.g., LinUCB (Abbasi-Yadkori et al., 2011)) which is common in real-world offline datasets and introduces dependencies, since a logging policy may violate the independency requirement.

**(2) Dependent actions with bounded variance.** Following early clustering works (Gentile et al., 2014; Li & Zhang, 2018; Li et al., 2019; Liu et al., 2022), this model allows dependencies among actions but requires an additional bounded-variance condition. This assumption still relies on finite candidate sets $\mathcal{S}_u^i$ but depends directly on $\lambda_a$ rather than $\tilde{\lambda}_a$. However, as argued in Wang et al. (2023b), bounded-variance assumptions may be unrealistically strong in practice and sometimes conflict with realistic data-generation processes.

**Why We Adopt Assumption 2.1 in the Main Body.** To remain consistent with state-of-the-art clustering of bandits work (Wang et al., 2023a; Dai et al., 2024a; Wang et al., 2025) and to better model realistic offline data (e.g., those generated by logging policies that induce dependencies or violate bounded-variance conditions), we adopt Assumption 2.1 in the main body. This assumption provides an interpretable and practical balance: it requires finite candidate sets (a natural fit for

recommendation systems and ranking tasks) and makes use of the smoothed parameter $\tilde{\lambda}_a$, which better captures the effective coverage of the action space under dependent samples. Importantly, our theoretical guarantees remain robust: they also hold under Assumption B.1, with only minor substitutions in the analysis. This illustrates the generality of our framework.

*Remark* B.2 (Beyond Regularity Assumptions). Finally, we consider scenarios where neither Assumption 2.1 nor Assumption B.1 holds. In our proofs (see Appendix D), the purpose of the regularity assumption is to ensure a lower bound on the minimum eigenvalue of the information matrix $M_u$ defined in Equation (1) (i.e. $\lambda_{\min}(M_u)$), so that the confidence interval $\text{CI}_u$ can be controlled in terms of $N_u$, the number of samples, as represented in Lemma E.2. Without regularity, the denominator of $\text{CI}_u$ must instead depend directly on $\sqrt{\lambda_{\min}(M_u)}$, as shown in Equation (11). In this case, all performance bounds would explicitly depend on $\lambda_{\min}(M_u)$ rather than sample size. While this generalization is conceptually straightforward, in order to keep consistent with the standard assumptions used throughout the clustering of bandits literature (Gentile et al., 2014; Li & Zhang, 2018; Li et al., 2019; Liu et al., 2022; Wang et al., 2023a; Dai et al., 2024a; Li et al., 2025; Wang et al., 2025), we still adopt Assumption 2.1 in the main body and leave detailed analysis of non-regular settings as a promising direction for future work.

## C ADDITIONAL THEORETICAL RESULTS

### C.1 FIXED-PARAMETER FORM FOR THEOREM 3.4

By refining the role of $\alpha_r$ and $\alpha_w$, we can derive an upper bound for Theorem 3.4 with fixed parameters. Specifically, we replace $N_{\mathcal{V}_{\hat{\gamma}}(u_{\text{test}})}$ with $N_{\mathcal{R}_{\hat{\gamma}}(u_{\text{test}})}$ in the denominator of the first term, select $\alpha_r$ at the lower end of its range (to conservatively estimate a subset of $\mathcal{R}_{\hat{\gamma}}(u_{\text{test}})$), and $\alpha_w$ at the upper end (to conservatively estimate a superset of $\mathcal{W}_{\hat{\gamma}}(u_{\text{test}})$). This yields the concrete form presented in Corollary C.1.

**Corollary C.1** (Concrete Form of Theorem 4.2). *Define $\tilde{\mathcal{R}}_{\hat{\gamma}}(u)$ and $\tilde{\mathcal{W}}_{\hat{\gamma}}(u)$ as the concrete instances of $\mathcal{R}_{\hat{\gamma}}(u)$ and $\mathcal{W}_{\hat{\gamma}}(u)$ in Lemma 3.2, obtained by setting $\alpha_r = \frac{\sqrt{\tilde{\lambda}_a}}{4(\alpha+1)\sqrt{\log(2U/\delta)\max\{2,d\}}}$ and $\alpha_w = \frac{\sqrt{\tilde{\lambda}_a}}{2(\alpha-1)\sqrt{\log(2U/\delta)}}$. Let $\tilde{\mathcal{V}}_{\hat{\gamma}}(u) = \tilde{\mathcal{R}}_{\hat{\gamma}}(u) \cup \tilde{\mathcal{W}}_{\hat{\gamma}}(u)$ and define $\eta_{\tilde{\mathcal{W}}_{\hat{\gamma}}(u)} = \frac{|\tilde{\mathcal{W}}_{\hat{\gamma}}(u)|\lambda + N_{\tilde{\mathcal{W}}_{\hat{\gamma}}(u)}}{|\tilde{\mathcal{V}}_{\hat{\gamma}}(u)|\lambda + N_{\tilde{\mathcal{V}}_{\hat{\gamma}}(u)}}$. Then, with probability at least $1 - 2\delta$, the suboptimality of Off-$C^2$LUB satisfies*

$$\text{SubOpt}(u_{\text{test}}, \text{Off-}C^2\text{LUB}, \mathcal{A}_{\text{test}}) \leq \tilde{O}\left(\sqrt{\frac{d}{\tilde{\lambda}_a N_{\tilde{\mathcal{R}}_{\hat{\gamma}}(u_{test})}}} + \frac{\eta_{\tilde{\mathcal{W}}_{\hat{\gamma}}(u_{test})}\hat{\gamma}}{\tilde{\lambda}_a}\right). \tag{7}$$

### C.2 ADDITIONAL DISCUSSIONS ON $\hat{\gamma}$ SELECTION POLICIES

**Theorem C.2** (Effect of Underestimation Policy). *With $\hat{\gamma}$ chosen according to the underestimation policy in Equation (5) and $\alpha'_w = \frac{\sqrt{\tilde{\lambda}_a}}{4(\alpha+1)\sqrt{\log(2U/\delta)\max\{2,d\}}}$, any user $v$ in the heterogeneous neighbor set $\mathcal{W}_{\hat{\gamma}}(u_{test})$ of Lemma 3.2 also satisfies $\frac{1}{\sqrt{N_{u_{test}}}} + \frac{1}{\sqrt{N_v}} \geq \alpha'_w \|\boldsymbol{\theta}_{u_{test}} - \boldsymbol{\theta}_v\|_2$.*

*Remark* C.3 (Interpretation of Underestimation Policy). Since $\Gamma(u, v) \leq \|\boldsymbol{\theta}_u - \boldsymbol{\theta}_v\|_2$, all users in $M(u_{\text{test}})$ must belong to clusters different from $u_{\text{test}}$. By taking the minimum over these values, the underestimation policy enforces a small $\hat{\gamma}$, which substantially limits the size of $\mathcal{W}_{\hat{\gamma}}(u_{\text{test}})$. This prevents the inclusion of highly biased heterogeneous samples, thereby reducing systematic bias in estimation. The additional sample-complexity condition in Theorem C.2 further guarantees that only heterogeneous users with insufficient data can enter $\mathcal{W}_{\hat{\gamma}}(u_{\text{test}})$, minimizing their overall impact. However, this conservative choice of $\hat{\gamma}$ also restricts the homogeneous set $\mathcal{R}_{\hat{\gamma}}(u_{\text{test}})$, leaving fewer samples available for aggregation and thus amplifying noise. The underestimation policy is therefore most suitable for safety-critical applications where bias is more damaging than noise, such as personalized medicine or risk-sensitive decision making.

**Theorem C.4** (Effect of Overestimation Policy). *With $\hat{\gamma}$ chosen according to the overestimation policy in Equation (5), if $M(u_{test}) \neq \emptyset$, then $\hat{\gamma} \geq \gamma$.*

*Remark* C.5 (Interpretation of Overestimation Policy). By construction, the overestimation policy ensures $\hat{\gamma} \geq \gamma$. This guarantees that more homogeneous users of $u_{\text{test}}$ with sufficient data will be included in $\mathcal{V}_{\hat{\gamma}}(u_{\text{test}})$ under the looser requirement of sample size as represented in Lemma 3.2. As a result, the aggregated dataset becomes significantly larger, which decreases variance and reduces noise in preference estimation. The drawback, however, is that this looser threshold also admits some heterogeneous users into $\mathcal{W}_{\hat{\gamma}}(u_{\text{test}})$, thereby introducing additional bias. The overestimation policy is thus well-suited for data-rich environments where variance dominates bias—for example, recommender systems with high-dimensional preference vectors, where aggregating large amounts of diverse data is essential for stable learning.

# D    DETAILED PROOFS

## D.1    PROOF OF LEMMA 3.2

We first define several events to bound the estimates. Let $\beta(\mathcal{S}, \delta) = \sqrt{d \log\left(1 + \frac{\sum_{u \in \mathcal{S}} N_u}{|\mathcal{S}|\lambda d}\right) + 2\log\left(\frac{2U}{\delta}\right)} + \sqrt{|\mathcal{S}|\lambda}$ for any set $\mathcal{S}$. We begin by defining the following confidence event for each user $u$:

$$\mathcal{F}'_u(\delta) = \left\{ \left\| \boldsymbol{\theta}_u - \hat{\boldsymbol{\theta}}_u \right\|_{M_u} \leq \beta(\{u\}, \delta) \right\},$$

where

$$M^{\mathcal{S}} = \lambda |\mathcal{S}| \cdot I + \sum_{u \in \mathcal{S}} \sum_{i=1}^{N_u} \boldsymbol{a}_u^i (\boldsymbol{a}_u^i)^\top, \quad \boldsymbol{b}^{\mathcal{S}} = \sum_{u \in \mathcal{S}} \sum_{i=1}^{N_u} r_u^i \boldsymbol{a}_u^i, \quad \boldsymbol{\theta}^{\mathcal{S}} = (M^{\mathcal{S}})^{-1} \boldsymbol{b}^{\mathcal{S}}.$$

By Lemma E.1, we have $\mathbb{P}[\mathcal{F}'_v(\delta)] \geq 1 - \frac{\delta}{2U}$ for each $v \in \mathcal{U}$. Therefore, applying the union bound over all users yields:

$$\mathbb{P}\left[ \bigcap_{v \in \mathcal{U}} \mathcal{F}'_v(\delta) \right] \geq 1 - \frac{\delta}{2}.$$

Furthermore, for both $\mathcal{R}_{\hat{\gamma}}(u)$ and $\mathcal{W}_{\hat{\gamma}}(u)$, the clustering condition in Equation (2) requires that a user $v$ can be included in either set only if both $N_u$ and $N_v$ are at least $\frac{16}{\tilde{\lambda}_a^2} \log\left(\frac{8dU}{\tilde{\lambda}_a^2 \delta}\right)$. Hence, under the case where $N_u < \frac{16}{\tilde{\lambda}_a^2} \log\left(\frac{8dU}{\tilde{\lambda}_a^2 \delta}\right)$, the sets $\mathcal{R}_{\hat{\gamma}}(u) = \{u\}$ and $\mathcal{W}_{\hat{\gamma}}(u) = \emptyset$ since the cluster condition in Equation (2) does not hold for any user $v$.

For the non-trivial case where $N_u \geq \frac{16}{\tilde{\lambda}_a^2} \log\left(\frac{8dU}{\tilde{\lambda}_a^2 \delta}\right)$, we apply Assumption 2.1, Lemma J.1 in Wang et al. (2023a), Lemma 7 in Li & Zhang (2018) and the union bound to obtain that for any $v \in \mathcal{R}_{\hat{\gamma}}(u) \cup \mathcal{W}_{\hat{\gamma}}(u)$, with probability at least $1 - \frac{3}{2}\delta$, the following inequality holds:

$$\left\| \boldsymbol{\theta}_v - \hat{\boldsymbol{\theta}}_v \right\|_2 \leq \frac{\left\| \boldsymbol{\theta}_v - \hat{\boldsymbol{\theta}}_v \right\|_{M_v}}{\sqrt{\lambda_{\min}(M_v)}} \leq \frac{\beta(\{v\}, \delta)}{\sqrt{\lambda + \tilde{\lambda}_a N_v / 2}} \leq \text{CI}_v.$$

Here, since $\hat{\gamma} \geq 0$, for any pair of connected users $(u_1, u_2)$, we have:

$$\begin{aligned}
\|\boldsymbol{\theta}_{u_1} - \boldsymbol{\theta}_{u_2}\|_2 &= \|\boldsymbol{\theta}_{u_1} - \hat{\boldsymbol{\theta}}_{u_1} + \hat{\boldsymbol{\theta}}_{u_1} - \hat{\boldsymbol{\theta}}_{u_2} + \hat{\boldsymbol{\theta}}_{u_2} - \boldsymbol{\theta}_{u_2}\|_2 \\
&\leq \|\boldsymbol{\theta}_{u_1} - \hat{\boldsymbol{\theta}}_{u_1}\|_2 + \|\hat{\boldsymbol{\theta}}_{u_1} - \hat{\boldsymbol{\theta}}_{u_2}\|_2 + \|\hat{\boldsymbol{\theta}}_{u_2} - \boldsymbol{\theta}_{u_2}\|_2 \\
&\leq \text{CI}_{u_1} + \|\hat{\boldsymbol{\theta}}_{u_1} - \hat{\boldsymbol{\theta}}_{u_2}\|_2 + \text{CI}_{u_2} \leq \hat{\gamma},
\end{aligned}$$

where the last inequality follows from the connection condition in Algorithm 1. Therefore, $\mathcal{R}_{\hat{\gamma}}(u)$ includes users that share the same preference vector as user $u$, while $\mathcal{W}_{\hat{\gamma}}(u)$ includes users whose preference vectors differ from that of $u$ by at most $\hat{\gamma} = \gamma + \varepsilon$.

Next, we prove Lemma 3.2, beginning with the representation of $\mathcal{R}_{\hat{\gamma}}(u)$ and then proceeding to that of $\mathcal{W}_{\hat{\gamma}}(u)$. Throughout, we focus on the case where $N_u \geq N_{\min}$.

**Proof of $\mathcal{R}_{\hat{\gamma}}(u)$.** We begin by analyzing the set $\mathcal{R}_{\hat{\gamma}}(u)$. Specifically, it suffices to show: With probability at least $1 - 2\delta$, (i) for $\alpha_{r1} = \frac{\sqrt{\tilde{\lambda}_a}}{4(\alpha+1)\sqrt{\log(2U/\delta)\max\{2,d\}}}$, any user $v$ with $\boldsymbol{\theta}_u = \boldsymbol{\theta}_v$, $1/\sqrt{N_u} + 1/\sqrt{N_v} \leq \alpha_{r1}\hat{\gamma}$, and $N_v \geq N_{\min}$ is included in $\mathcal{R}_{\hat{\gamma}}(u)$ when $N_u \geq N_{\min}$; (ii) for $\alpha_{r2} = \frac{\sqrt{\tilde{\lambda}_a}}{2\alpha\sqrt{\log(2U/\delta)}}$, any user $v$ with $\boldsymbol{\theta}_u \neq \boldsymbol{\theta}_v$, or $1/\sqrt{N_u} + 1/\sqrt{N_v} > \alpha_{r2}\hat{\gamma}$, or $N_v < N_{\min}$ cannot be included in $\mathcal{R}_{\hat{\gamma}}(u)$.

*Proof of (i).* Note that the condition $1/\sqrt{N_u} + 1/\sqrt{N_v} \leq \alpha_{r1}\hat{\gamma}$ implies

$$(\alpha+1)(\mathrm{CI}_u + \mathrm{CI}_v) \leq \frac{2(\alpha+1)\sqrt{4\max\{2,d\}\log(\frac{2U}{\delta})}}{\sqrt{\tilde{\lambda}_a N_u}} + \frac{2(\alpha+1)\sqrt{4\max\{2,d\}\log(\frac{2U}{\delta})}}{\sqrt{\tilde{\lambda}_a N_v}}$$

$$< \hat{\gamma},$$

under the parameter selections for $\lambda$ and $\delta$ such that

$$\lambda \leq d\log\left(1 + \frac{\min_w\{N_w\}}{\lambda d}\right) + 2\log\left(\frac{2U}{\delta}\right), \quad \log\left(1 + \frac{\max_w\{N_w\}}{\lambda d}\right) \leq \log\left(\frac{2U}{\delta}\right).$$

Given that $\boldsymbol{\theta}_u = \boldsymbol{\theta}_v$, we further observe:

$$\hat{\gamma} > (\alpha+1)(\mathrm{CI}_u + \mathrm{CI}_v)$$
$$= \alpha(\mathrm{CI}_u + \mathrm{CI}_v) + \|\boldsymbol{\theta}_u - \boldsymbol{\theta}_v\|_2 + \mathrm{CI}_u + \mathrm{CI}_v$$
$$\geq \|\hat{\boldsymbol{\theta}}_u - \hat{\boldsymbol{\theta}}_v\|_2 + \alpha(\mathrm{CI}_u + \mathrm{CI}_v),$$

with probability at least $1 - 2\delta$. This directly implies that the connection condition in Algorithm 1 is satisfied, and therefore user $v$ is included in $\mathcal{R}_{\hat{\gamma}}(u)$.

*Proof of (ii).* If user $v$ satisfies either $\boldsymbol{\theta}_u \neq \boldsymbol{\theta}_v$ or $N_v < N_{\min}$, it is straightforward that $v \notin \mathcal{R}_{\hat{\gamma}}(u)$. We therefore focus on showing that the condition $1/\sqrt{N_u} + 1/\sqrt{N_v} > \alpha_{r2}\hat{\gamma}$ also implies $v \notin \mathcal{R}_{\hat{\gamma}}(u)$.

Note that the condition that $1/\sqrt{N_u} + 1/\sqrt{N_v} > \alpha_{r2}\hat{\gamma}$ implies

$$\alpha(\mathrm{CI}_u + \mathrm{CI}_v) > 2\alpha\left(\sqrt{\frac{\log\left(\frac{2U}{\delta}\right)}{\tilde{\lambda}_a N_u}} + \sqrt{\frac{\log\left(\frac{2U}{\delta}\right)}{\tilde{\lambda}_a N_v}}\right) > \hat{\gamma}.$$

Hence the connecting condition in Equation (2) does not hold due to the positive value of $\left\|\hat{\boldsymbol{\theta}}_{u_1} - \hat{\boldsymbol{\theta}}_{u_2}\right\|_2$ and $v \notin \mathcal{R}_{\hat{\gamma}}(u)$ holds.

**Proof of $\mathcal{W}_{\hat{\gamma}}(u)$.** First it is trivial that $\alpha_w > 0$ holds. As we have proved that for any $v \in \mathcal{W}_{\hat{\gamma}}(u)$ it holds that $\gamma \leq \|\boldsymbol{\theta}_u - \boldsymbol{\theta}_v\|_2 \leq \hat{\gamma}$, therefore it suffices to show that if $1/\sqrt{N_u} + 1/\sqrt{N_v} > \frac{\varepsilon\sqrt{\tilde{\lambda}_a}}{2(\alpha-1)\sqrt{\log(2U/\delta)}}$ and $\|\boldsymbol{\theta}_u - \boldsymbol{\theta}_v\|_2 \geq \gamma$, then user $v$ is not included in $\mathcal{W}_{\hat{\gamma}}(u)$ with probability at least $1 - 2\delta$ when $\min\{N_u, N_v\} \geq N_{\min}$.

Observe that the condition $1/\sqrt{N_u} + 1/\sqrt{N_v} > \frac{\varepsilon\sqrt{\tilde{\lambda}_a}}{2(\alpha-1)\sqrt{\log(2U/\delta)}}$ implies:

$$(\alpha-1)(\mathrm{CI}_u + \mathrm{CI}_v) \geq (\alpha-1)\left(\sqrt{\frac{4\log\left(\frac{2U}{\delta}\right)}{\tilde{\lambda}_a N_u}} + \sqrt{\frac{4\log\left(\frac{2U}{\delta}\right)}{\tilde{\lambda}_a N_v}}\right)$$

$$> \varepsilon \geq \hat{\gamma} - \|\boldsymbol{\theta}_u - \boldsymbol{\theta}_v\|_2.$$

Therefore, we can deduce that:

$$\left\|\hat{\boldsymbol{\theta}}_u - \hat{\boldsymbol{\theta}}_v\right\|_2 + \alpha(\mathrm{CI}_u + \mathrm{CI}_v) \geq \alpha(\mathrm{CI}_u + \mathrm{CI}_v) + \|\boldsymbol{\theta}_u - \boldsymbol{\theta}_v\|_2 - \mathrm{CI}_u - \mathrm{CI}_v$$

$$> \hat{\gamma},$$

which means that the connection condition in Equation (2) does not hold. Consequently, user $v$ is not connected to $u$, and therefore not included in $\mathcal{W}_{\hat{\gamma}}(u)$ with probability at least $1 - 2\delta$, provided that $\alpha_w \in \left(0, \frac{\sqrt{\tilde{\lambda}_a}}{2(\alpha-1)\sqrt{\log(2U/\delta)}}\right)$.

## D.2 PROOF OF THEOREM 3.4

*Proof of Theorem 3.4.* To prove Theorem 3.4, we first recall and several key notations and introduce some new notations in Table 5.

Table 5: Notations in Proofs

| Notation | Description |
|---|---|
| $\mathcal{V}_{\hat{\gamma}}(u)$ | Set consisting of user $u$ and its neighbors in the graph $\mathcal{G}_{\hat{\gamma}}$. |
| $\mathcal{R}_{\hat{\gamma}}(u)$ | Set of users in $\mathcal{V}_{\hat{\gamma}}(u)$ that share the same preference vector as user $u$. |
| $\mathcal{W}_{\hat{\gamma}}(u)$ | Set of users in $\mathcal{V}_{\hat{\gamma}}(u)$ that do *not* share the same preference vector as user $u$. |
| $\mathcal{W}_{\hat{\gamma}}^j(u)$ | Set of users in $\mathcal{W}_{\hat{\gamma}}(u)$ who belong to cluster $\mathcal{V}(j)$. |
| $\mathcal{J}_{\hat{\gamma}}(u)$ | Set of indices of distinct clusters represented in $\mathcal{W}_{\hat{\gamma}}(u)$. |

To characterize the contributions of different types of neighbors in the estimation process, we define the weighted fraction of samples from $\mathcal{R}_{\hat{\gamma}}(u)$ relative to $\mathcal{V}_{\hat{\gamma}}(u)$ as:

$$\eta_{\mathcal{R}_{\hat{\gamma}}(u)} = \frac{|\mathcal{R}_{\hat{\gamma}}(u)|\lambda + N_{\mathcal{R}_{\hat{\gamma}}(u)}}{|\mathcal{V}_{\hat{\gamma}}(u)|\lambda + N_{\mathcal{V}_{\hat{\gamma}}(u)}}.$$

Here, $\lambda$ is the regularization parameter used in constructing each user's matrix $M_u$. Analogous definitions apply to $\eta_{\mathcal{W}_{\hat{\gamma}}(u)}$ and $\eta_{\mathcal{W}_{\hat{\gamma}}^j(u)}$, and by construction, we have:

$$\eta_{\mathcal{R}_{\hat{\gamma}}(u)} + \eta_{\mathcal{W}_{\hat{\gamma}}(u)} = 1,$$

since $\mathcal{V}_{\hat{\gamma}}(u) = \mathcal{R}_{\hat{\gamma}}(u) \cup \mathcal{W}_{\hat{\gamma}}(u)$ and the two sets are disjoint.

We now turn to bounding the suboptimality term $\mathrm{SubOpt}(u, \pi, \mathcal{A}_{\text{test}}) = \langle \boldsymbol{\theta}_u, \boldsymbol{a}_u^* \rangle - \langle \boldsymbol{\theta}_u, \boldsymbol{a}_{\text{test}} \rangle$. Firstly, the case where $N_u < N_{\min}$ is trivial since the problem can be simplified to an offline linear contextual bandit problem in (Li et al., 2022) as the algorithm only makes use of the data from user $u$ itself. For the non-trivial case where $N_u \geq N_{\min}$, recall that in Appendix D.1 we proved that $\left\| \boldsymbol{\theta}_v - \hat{\boldsymbol{\theta}}_v \right\|_2 \leq \mathrm{CI}_v$ for any user $v$ with probability at least $1 - \frac{3}{2}\delta$, and $\|\boldsymbol{\theta}_{u_1} - \boldsymbol{\theta}_{u_2}\|_2 \leq \hat{\gamma}$ holds for any connected users $(u_1, u_2)$.

For notational simplicity, we let:

$$V = \mathcal{V}_{\hat{\gamma}}(u), \quad R = \mathcal{R}_{\hat{\gamma}}(u), \quad W = \mathcal{W}_{\hat{\gamma}}(u), \quad W^j = \mathcal{W}_{\hat{\gamma}}^j(u), \quad \tilde{J} = \mathcal{J}_{\hat{\gamma}}(u).$$

Moreover, for a user set $\mathcal{S}$, we define the following aggregated statistics:

$$M^{\mathcal{S}} = \lambda|\mathcal{S}|I + \sum_{u \in \mathcal{S}} \sum_{i=1}^{N_u} \boldsymbol{a}_u^i (\boldsymbol{a}_u^i)^\top, \quad \boldsymbol{b}^{\mathcal{S}} = \sum_{u \in \mathcal{S}} \sum_{i=1}^{N_u} r_u^i \boldsymbol{a}_u^i, \quad \hat{\boldsymbol{\theta}}^{\mathcal{S}} = (M^{\mathcal{S}})^{-1} \boldsymbol{b}^{\mathcal{S}}.$$

Recall that $\boldsymbol{b}^V = \boldsymbol{b}^R + \sum_{j \in \tilde{J}} \boldsymbol{b}^{W^j}$ and $M^V = M^R + \sum_{j \in \tilde{J}} M^{W^j}$, we can decompose $\hat{\boldsymbol{\theta}}^V$ as:

$$\hat{\boldsymbol{\theta}}^V = (M^V)^{-1} \boldsymbol{b}^V = (M^V)^{-1}\left(\boldsymbol{b}^R + \sum_{j \in \tilde{J}} \boldsymbol{b}^{W^j}\right)$$

$$= (M^V)^{-1} M^R (M^R)^{-1} \boldsymbol{b}^R + \sum_{j \in \tilde{J}} (M^V)^{-1} M^{W^j} (M^{W^j})^{-1} \boldsymbol{b}^{W^j}$$

$$= (M^V)^{-1} M^R \hat{\boldsymbol{\theta}}^R + \sum_{j \in \tilde{J}} (M^V)^{-1} M^{W^j} \hat{\boldsymbol{\theta}}^{W^j}.$$

Let $\varepsilon_v^i = r_v^i - \boldsymbol{\theta}_v^\top \boldsymbol{a}_v^i$ denote the error for the sample $(\boldsymbol{a}_v^i, r_v^i) \in \mathcal{D}_v$ where $i \in [N_v]$. Hence, we have:

$$\left\langle \hat{\boldsymbol{\theta}}^V - \boldsymbol{\theta}_u, \boldsymbol{a}_{\text{test}} \right\rangle = \left\langle (M^V)^{-1} M^R \hat{\boldsymbol{\theta}}^R + \sum_{j \in \tilde{J}} (M^V)^{-1} M^{W^j} \hat{\boldsymbol{\theta}}^{W^j} - \boldsymbol{\theta}_u, \boldsymbol{a}_{\text{test}} \right\rangle \quad (8)$$

$$= \left\langle (M^V)^{-1} M^R (\hat{\boldsymbol{\theta}}^R - \boldsymbol{\theta}_u) + \sum_{j \in \tilde{J}} (M^V)^{-1} M^{W^j} (\hat{\boldsymbol{\theta}}^{W^j} - \boldsymbol{\theta}_u), \boldsymbol{a}_{\text{test}} \right\rangle$$

$$= \left\langle (M^V)^{-1} M^R (\hat{\boldsymbol{\theta}}^R - \boldsymbol{\theta}_u), \boldsymbol{a}_{\text{test}} \right\rangle + \left\langle \sum_{j \in \tilde{J}} (M^V)^{-1} M^{W^j} (\hat{\boldsymbol{\theta}}^{W^j} - \boldsymbol{\theta}_u), \boldsymbol{a}_{\text{test}} \right\rangle$$

$$= \left\langle (M^V)^{-1} \left( \boldsymbol{b}^R - M^R \boldsymbol{\theta}_u + \sum_{j \in \tilde{J}} (\boldsymbol{b}^{W^j} - M^{W^j} \boldsymbol{\theta}^j) \right), \boldsymbol{a}_{\text{test}} \right\rangle$$

$$+ \left\langle \sum_{j \in \tilde{J}} (M^V)^{-1} M^{W^j} (\boldsymbol{\theta}^j - \boldsymbol{\theta}_u), \boldsymbol{a}_{\text{test}} \right\rangle$$

$$= \left\langle (M^V)^{-1} \sum_{v \in V} \sum_{i=1}^{N_v} \varepsilon_v^i \boldsymbol{a}_v^i, \boldsymbol{a}_{\text{test}} \right\rangle - \left\langle (M^V)^{-1} \lambda \left( |R| \boldsymbol{\theta}_u + \sum_{j \in \tilde{J}} |W^j| \boldsymbol{\theta}^j \right), \boldsymbol{a}_{\text{test}} \right\rangle$$

$$+ \left\langle \sum_{j \in \tilde{J}} (M^V)^{-1} M^{W^j} (\boldsymbol{\theta}^j - \boldsymbol{\theta}_u), \boldsymbol{a}_{\text{test}} \right\rangle \tag{9}$$

Applying the Cauchy–Schwarz inequality, we obtain from Equation (9):

$$\left\langle \hat{\boldsymbol{\theta}}^V - \boldsymbol{\theta}_u, \boldsymbol{a}_{\text{test}} \right\rangle \leq \left\| (M^V)^{-1/2} \boldsymbol{a}_{\text{test}} \right\|_2 \cdot \left\| \sum_{v \in V} \sum_{i=1}^{N_v} \varepsilon_v^i \boldsymbol{a}_v^i \right\|_{(M^V)^{-1}}$$

$$+ \left\| \lambda (M^V)^{-1} \left( |R| \boldsymbol{\theta}_u + \sum_{j \in \tilde{J}} |W^j| \boldsymbol{\theta}^j \right) \right\|_2 \cdot \| \boldsymbol{a}_{\text{test}} \|_2$$

$$+ \left\| \sum_{j \in \tilde{J}} (M^V)^{-1} M^{W^j} (\boldsymbol{\theta}^j - \boldsymbol{\theta}_u) \right\|_2 \cdot \| \boldsymbol{a}_{\text{test}} \|_2 \tag{10}$$

We now bound each term using matrix norm properties and known results from Equation (10):

$$\left\langle \hat{\boldsymbol{\theta}}^V - \boldsymbol{\theta}_u, \boldsymbol{a}_{\text{test}} \right\rangle \leq \frac{\beta(V, \delta)}{\sqrt{\lambda_{\min}(V)}} + \frac{\lambda |V|}{\lambda_{\min}(V)} + \frac{\lambda_{\max}(W)}{\lambda_{\min}(V)} \cdot \hat{\gamma} \tag{11}$$

$$\leq \frac{\sqrt{2} \beta(V, \delta)}{\sqrt{\tilde{\lambda}_a N_V}} + \frac{2\lambda U}{\tilde{\lambda}_a N_V} + \frac{2\eta_W}{\tilde{\lambda}_a} \cdot \hat{\gamma} =: \Psi \tag{12}$$

with probability at least $1 - 2\delta$.

Here, Equation (8) follows from the decomposition $M^V = M^R + \sum_{j \in \tilde{J}} M^{W^j}$; Equation (9) uses the fact that $M^S = \lambda |S| I + \sum_{v \in S} \sum_{i=1}^{N_v} \boldsymbol{a}_v^i (\boldsymbol{a}_v^i)^\top$ and the definition of reward noise $\varepsilon_v^i$. Equation (10) is due to the Cauchy–Schwarz inequality. The first inequality in Equation (12) uses eigenvalue bounds and Theorem 1 in (Abbasi-Yadkori et al., 2011), while the second inequality follows from bounding eigenvalues via the number of samples and applying Assumption 2.1, Lemma J.1 in Wang et al. (2023a) and Lemma 7 in Li & Zhang (2018).

Therefore, we obtain:

$$-\langle \boldsymbol{\theta}_u, \boldsymbol{a}_{\text{test}} \rangle \leq \Psi - \left\langle \hat{\boldsymbol{\theta}}^V, \boldsymbol{a}_{\text{test}} \right\rangle.$$

Recall that the selected action $\boldsymbol{a}_{\text{test}}$ satisfies the confidence bound:

$$\left( \hat{\boldsymbol{\theta}}^V \right)^\top \boldsymbol{a}_{\text{test}} - \beta(V, \delta) \| \boldsymbol{a}_{\text{test}} \|_{(M^V)^{-1}} \geq \left( \hat{\boldsymbol{\theta}}^V \right)^\top \boldsymbol{a}_u^* - \beta(V, \delta) \| \boldsymbol{a}_u^* \|_{(M^V)^{-1}}.$$

Hence, we can derive the following bound for the suboptimality:

$$\text{SubOpt}(u, \pi, \mathcal{A}_{\text{test}}) = \langle \boldsymbol{\theta}_u, \boldsymbol{a}_u^* \rangle - \langle \boldsymbol{\theta}_u, \boldsymbol{a}_{\text{test}} \rangle$$

$$\leq \langle \boldsymbol{\theta}_u, \boldsymbol{a}_u^* \rangle - \left\langle \hat{\boldsymbol{\theta}}^V, \boldsymbol{a}_{\text{test}} \right\rangle + \Psi$$

$$\leq \left\langle \boldsymbol{\theta}_u - \hat{\boldsymbol{\theta}}^V, \boldsymbol{a}_u^* \right\rangle + \beta(V, \delta) \left( \|\boldsymbol{a}_u^*\|_{(M^V)^{-1}} - \|\boldsymbol{a}_{\text{test}}\|_{(M^V)^{-1}} \right) + \Psi$$

$$\leq 2\Psi + \frac{\sqrt{2}\beta(V, \delta)}{\sqrt{\lambda_{\min}(M^V)}}$$

$$\leq 2\Psi + \frac{\beta(V, \delta)}{\sqrt{\tilde{\lambda}_a N_V}}$$

with probability at least $1 - 2\delta$. By omitting the logarithmic and constant terms in the above inequality we complete the proof of Theorem 3.4. □

### D.3 PROOF OF LEMMA C.2

*Proof of Lemma C.2.* Recall that we define

$$\hat{\gamma} = \mathbb{I}\{M(u_{\text{test}}) \neq \emptyset\} \cdot \min_{v \in M(u_{\text{test}})} \Gamma(u_{\text{test}}, v), \tag{13}$$

where $M(u_{\text{test}})$ is the set of users for which the computed value $\Gamma(u_{\text{test}}, v)$ is positive.

Let $s = \arg\min_{v \neq u_{\text{test}}} \{\Gamma(u_{\text{test}}, v) > 0\}$, which corresponds to the user achieving the minimum positive value and hence defines $\hat{\gamma}$. Then, for any other user $v$, we must have either $\Gamma(u_{\text{test}}, v) \leq 0$ or $\Gamma(u_{\text{test}}, v) \geq \hat{\gamma}$.

We will show that any user included in $\mathcal{W}_{\hat{\gamma}}(u_{\text{test}})$, as represented in Lemma 3.2, also satisfies the condition in Lemma C.2. For simplicity, we denote $u = u_{\text{test}}$ in the rest of the proof.

We first consider the case where $\Gamma(u, v) \leq 0$. This implies:

$$\left\| \hat{\boldsymbol{\theta}}_u - \hat{\boldsymbol{\theta}}_v \right\|_2 - \alpha(\text{CI}_u + \text{CI}_v) \leq 0. \tag{14}$$

By the triangle inequality, we have:

$$\|\boldsymbol{\theta}_u - \boldsymbol{\theta}_v\|_2 \leq \left\| \hat{\boldsymbol{\theta}}_u - \hat{\boldsymbol{\theta}}_v \right\|_2 + \|\hat{\boldsymbol{\theta}}_u - \boldsymbol{\theta}_u\|_2 + \|\hat{\boldsymbol{\theta}}_v - \boldsymbol{\theta}_v\|_2$$

$$\leq \left\| \hat{\boldsymbol{\theta}}_u - \hat{\boldsymbol{\theta}}_v \right\|_2 + \text{CI}_u + \text{CI}_v$$

$$\leq (\alpha + 1)(\text{CI}_u + \text{CI}_v).$$

Furthermore, using the definition of $\text{CI}_u$ and $\text{CI}_v$, we obtain:

$$\|\boldsymbol{\theta}_u - \boldsymbol{\theta}_v\|_2 \leq \frac{4(\alpha + 1)\sqrt{\log(2U/\delta)\max\{2, d\}}}{\sqrt{\tilde{\lambda}_a}} \left( \frac{1}{\sqrt{N_u}} + \frac{1}{\sqrt{N_v}} \right).$$

This implies that

$$\frac{1}{\sqrt{N_u}} + \frac{1}{\sqrt{N_v}} \geq \frac{\sqrt{\tilde{\lambda}_a}}{4(\alpha + 1)\sqrt{\log(2U/\delta)\max\{2, d\}}} \|\boldsymbol{\theta}_u - \boldsymbol{\theta}_v\|_2.$$

Thus, by letting $\alpha'_w = \frac{\sqrt{\tilde{\lambda}_a}}{4(\alpha+1)\sqrt{\max\{2,d\}\log(2U/\delta)}}$, we obtain the desired result:

$$\frac{1}{\sqrt{N_u}} + \frac{1}{\sqrt{N_v}} \geq \alpha'_w \|\boldsymbol{\theta}_u - \boldsymbol{\theta}_v\|_2.$$

Then for the second case that $\Gamma(u_{\text{test}}, v) \geq \hat{\gamma}$, it can be proved that

$$\|\boldsymbol{\theta}_u - \boldsymbol{\theta}_v\|_2 \geq \left\| \hat{\boldsymbol{\theta}}_u - \hat{\boldsymbol{\theta}}_v \right\|_2 - \alpha(\text{CI}_u + \text{CI}_v) \geq \hat{\gamma}.$$

Thus $v \notin \mathcal{W}_{\hat{\gamma}}(u)$ holds. This ends the proof of Theorem C.2. □

### D.4 PROOF OF LEMMA C.4

*Proof of Lemma C.4.* The proof of Lemma C.4 follows from the fact that for all $v \in M(u_{\text{test}})$, as $M(u_{\text{test}})$ denotes the users belong to different clusters of $u_{\text{test}}$. By combining Equation (4), we have

$$\tilde{\Gamma}(u_{\text{test}}, v) \geq \|\boldsymbol{\theta}_{u_{\text{test}}} - \boldsymbol{\theta}_v\|_2 \geq \gamma.$$

Therefore, as long as $M(u_{\text{test}}) \neq \emptyset$, the optimistic estimation ensures that $\hat{\gamma}$ is at least $\gamma$. $\qquad\square$

### D.5 PROOF OF THEOREM 4.2

*Proof of Theorem 4.2.* Let $\beta(n, \delta) = \sqrt{d \log\left(1 + \frac{n}{\lambda d}\right) + 2 \log\left(\frac{1}{\delta}\right)} + \sqrt{\lambda}$, and let $\tilde{j}_u$ denote the cluster index of user $u$ in the clustering output $\tilde{\mathcal{G}}$. We denote by $\tilde{\mathcal{V}}(j)$ the $j$-th cluster in $\tilde{\mathcal{G}}$.

We first introduce the following events:

$$\mathcal{E} = \left\{\text{All clusters are correctly identified, i.e., } \mathcal{V}(j_u) = \tilde{\mathcal{V}}(\tilde{j}_u) \text{ for all } u \in \mathcal{V}\right\},$$

$$\mathcal{F}_u(\delta) = \left\{\left\|\boldsymbol{\theta}_u - \tilde{\boldsymbol{\theta}}_u\right\|_{\tilde{M}_u} \leq \beta(\tilde{N}_u, \delta)\right\}, \quad \forall u \in \mathcal{V},$$

$$\mathcal{F}'_u(\delta) = \left\{\left\|\boldsymbol{\theta}_u - \hat{\boldsymbol{\theta}}_u\right\|_{M_u} \leq \beta(N_u, \delta)\right\}, \quad \forall u \in \mathcal{V}. \tag{15}$$

We first verify that event $\mathcal{E}$ holds with high probability. According to Lemma E.1, the event $\mathcal{F}'_u(\delta/(2U))$ holds for all $u \in \mathcal{V}$ with probability at least $1 - \frac{1}{2}\delta$. Under this event, we have:

$$\begin{aligned}
\left\|\hat{\boldsymbol{\theta}}_u - \boldsymbol{\theta}_u\right\|_2 &\leq \frac{\left\|\hat{\boldsymbol{\theta}}_u - \boldsymbol{\theta}_u\right\|_{M_u}}{\sqrt{\lambda_{\min}(M_u)}} \\
&\leq \frac{\beta(N_u, \delta/(2U))}{\sqrt{\lambda + \lambda_{\min}\left(\sum_{i=1}^{N_u} \boldsymbol{a}_u^i (\boldsymbol{a}_u^i)^\top\right)}} \\
&\leq \frac{\sqrt{d \log\left(1 + \frac{N_u}{\lambda d}\right) + 2 \log\left(\frac{2U}{\delta}\right)} + \sqrt{\lambda}}{\sqrt{\lambda + \tilde{\lambda}_a N_u/2}} \\
&\leq \alpha \cdot \mathrm{CI}_u \leq \frac{\gamma}{4}, \tag{16}
\end{aligned}$$

with probability at least $1 - \frac{3}{2}\delta$, for any $u$ such that

$$N_u \geq \max\left\{\frac{512d}{\gamma^2 \tilde{\lambda}_a} \log\left(\frac{2U}{\delta}\right), \frac{16}{\tilde{\lambda}_a^2} \log\left(\frac{8dU}{\tilde{\lambda}_a^2 \delta}\right)\right\}.$$

The last inequality in Equation (16) follows from Lemma E.5.

Now consider the connection rule in Algorithm 2, specifically the separation condition in Equation (6). For any users $u_1, u_2 \in \mathcal{V}$, the algorithm distinguishes them into different clusters if

$$\alpha(\mathrm{CI}_{u_1} + \mathrm{CI}_{u_2}) < \left\|\hat{\boldsymbol{\theta}}_{u_1} - \hat{\boldsymbol{\theta}}_{u_2}\right\|_2.$$

From the triangle inequality, we have:

$$\begin{aligned}
\left\|\hat{\boldsymbol{\theta}}_{u_1} - \hat{\boldsymbol{\theta}}_{u_2}\right\|_2 &\leq \|\boldsymbol{\theta}_{u_1} - \boldsymbol{\theta}_{u_2}\|_2 + \left\|\hat{\boldsymbol{\theta}}_{u_1} - \boldsymbol{\theta}_{u_1}\right\|_2 + \left\|\hat{\boldsymbol{\theta}}_{u_2} - \boldsymbol{\theta}_{u_2}\right\|_2 \\
&\leq \|\boldsymbol{\theta}_{u_1} - \boldsymbol{\theta}_{u_2}\|_2 + \alpha \cdot \mathrm{CI}_{u_1} + \alpha \cdot \mathrm{CI}_{u_2}.
\end{aligned}$$

Therefore, combining both bounds gives:

$$\alpha(\mathrm{CI}_{u_1} + \mathrm{CI}_{u_2}) < \|\boldsymbol{\theta}_{u_1} - \boldsymbol{\theta}_{u_2}\|_2 + \alpha(\mathrm{CI}_{u_1} + \mathrm{CI}_{u_2}),$$

which implies $\|\boldsymbol{\theta}_{u_1} - \boldsymbol{\theta}_{u_2}\|_2 > 0$, and since the true cluster gap is at least $\gamma$, we conclude:

$$\|\boldsymbol{\theta}_{u_1} - \boldsymbol{\theta}_{u_2}\|_2 \geq \gamma,$$

i.e., $u_1$ and $u_2$ belong to different clusters. This confirms that the algorithm correctly separates users from different clusters with high probability, and hence event $\mathcal{E}$ holds with probability at least $1 - \frac{3}{2}\delta$.

On the other hand, for users $u_1, u_2$ that do not satisfy the separation condition, we have:

$$\gamma \geq 2\alpha(\mathrm{CI}_{u_1} + \mathrm{CI}_{u_2}) \geq \left\|\hat{\boldsymbol{\theta}}_{u_1} - \hat{\boldsymbol{\theta}}_{u_2}\right\|_2 + \alpha(\mathrm{CI}_{u_1} + \mathrm{CI}_{u_2})$$

$$\geq \left\|\boldsymbol{\theta}_{u_1} - \hat{\boldsymbol{\theta}}_{u_1}\right\|_2 + \left\|\hat{\boldsymbol{\theta}}_{u_1} - \hat{\boldsymbol{\theta}}_{u_2}\right\|_2 + \left\|\boldsymbol{\theta}_{u_2} - \hat{\boldsymbol{\theta}}_{u_2}\right\|_2 \geq \|\boldsymbol{\theta}_{u_1} - \boldsymbol{\theta}_{u_2}\|_2.$$

This implies that $u_1$ and $u_2$ must belong to the same cluster. Therefore, the clustering result satisfies event $\mathcal{E}$ with probability at least $1 - \delta$.

Furthermore, due to the homogeneity of users within each cluster $\tilde{\mathcal{V}}(j)$, the event $\mathcal{F}'_u(\delta/(2U))$ also holds for all $u \in \mathcal{V}$ with probability at least $1 - \delta$.

We now analyze the suboptimality of Off-CLUB. Recall that:

$$\mathrm{SubOpt}(u_{\text{test}}, \text{Off-CLUB}, A_{\text{test}}) = \langle \boldsymbol{\theta}_u, \boldsymbol{a}^*_{\text{test}} \rangle - \langle \boldsymbol{\theta}_u, \boldsymbol{a}_{\text{test}} \rangle.$$

When event $\mathcal{F}_u(\delta/(2U))$ holds for all $u \in \mathcal{V}$, by the Cauchy–Schwarz inequality we have:

$$\left\langle \tilde{\boldsymbol{\theta}}_u - \boldsymbol{\theta}_u, \boldsymbol{a}_{\text{test}} \right\rangle \leq \beta\left(\tilde{N}_u, \delta/(2U)\right) \cdot \|\boldsymbol{a}_{\text{test}}\|_{\tilde{M}_u^{-1}}.$$

Therefore,

$$- \langle \boldsymbol{\theta}_u, \boldsymbol{a}_{\text{test}} \rangle \leq - \left\langle \tilde{\boldsymbol{\theta}}_u, \boldsymbol{a}_{\text{test}} \right\rangle + \beta\left(\tilde{N}_u, \delta/(2U)\right) \cdot \|\boldsymbol{a}_{\text{test}}\|_{\tilde{M}_u^{-1}}$$

$$\leq - \left\langle \tilde{\boldsymbol{\theta}}_u, \boldsymbol{a}^*_{\text{test}} \right\rangle + \beta\left(\tilde{N}_u, \delta/(2U)\right) \cdot \|\boldsymbol{a}^*_{\text{test}}\|_{\tilde{M}_u^{-1}}. \tag{17}$$

Then, it follows that

$$\mathrm{SubOpt}(u, \pi, \mathcal{A}_{\text{test}}) \leq \left(\boldsymbol{\theta}_u - \tilde{\boldsymbol{\theta}}_u\right)^\top \boldsymbol{a}^*_{\text{test}} + \beta\left(\tilde{N}_u, \delta/(2U)\right) \cdot \|\boldsymbol{a}^*_{\text{test}}\|_{\tilde{M}_u^{-1}}$$

$$\leq \|\boldsymbol{a}^*_{\text{test}}\|_{\tilde{M}_u^{-1}} \cdot \left\|\tilde{\boldsymbol{\theta}}_u - \boldsymbol{\theta}_u\right\|_{\tilde{M}_u} + \beta\left(\tilde{N}_u, \delta/(2U)\right) \cdot \|\boldsymbol{a}^*_{\text{test}}\|_{\tilde{M}_u^{-1}}$$

$$\leq 2 \cdot \beta\left(\tilde{N}_u, \delta/(2U)\right) \cdot \|\boldsymbol{a}^*_{\text{test}}\|_{\tilde{M}_u^{-1}},$$

where the second inequality uses Cauchy–Schwarz, and the last follows from the definition of event $\mathcal{F}_u(\delta/(2U))$.

Note that

$$\|\boldsymbol{a}^*_{\text{test}}\|_{\tilde{M}_u^{-1}} = \sqrt{(\boldsymbol{a}^*_{\text{test}})^\top \tilde{M}_u^{-1} \boldsymbol{a}^*_{\text{test}}} = \sqrt{\left\langle \tilde{M}_u^{-1}, \boldsymbol{a}^*_{\text{test}} (\boldsymbol{a}^*_{\text{test}})^\top \right\rangle},$$

and by definition $\tilde{M}_u = \lambda I + \sum_{(\boldsymbol{a},r) \in \tilde{\mathcal{D}}_u} \boldsymbol{a}\boldsymbol{a}^\top$.

From Assumption 2.1, Lemma J.1 in Wang et al. (2023a) and Lemma 7 in Li & Zhang (2018), we know

$$\lambda_{\min}(\tilde{M}_u) \geq \lambda + \frac{\tilde{\lambda}_a N_{\mathcal{V}(j_u)}}{2},$$

and thus

$$\|\boldsymbol{a}^*_{\text{test}}\|_{\tilde{M}_u^{-1}} \leq \frac{1}{\sqrt{\lambda + \tilde{\lambda}_a N_{\mathcal{V}(j_u)}/2}}.$$

Combining this with the definition of $\beta$, we conclude:

$$\text{SubOpt}(u_{\text{test}}, \text{Off-CLUB}, \mathcal{A}_{\text{test}}) \leq \frac{\sqrt{d \log\left(1 + \frac{N_{\mathcal{V}(j_{u_{\text{test}}})}}{\lambda d}\right) + 2 \log\left(\frac{2U}{\delta}\right)} + \sqrt{\lambda}}{\sqrt{\lambda + \tilde{\lambda}_a N_{\mathcal{V}(j_{u_{\text{test}}})}/2}}, \qquad (18)$$

which holds with probability at least $1 - 2\delta$. Using the order $\tilde{O}$ to denote Equation (18) ends our proof. $\qquad\square$

### D.6 PROOF OF THEOREM 4.4

*Proof of Theorem 4.4.* Note that within the dataset $\mathcal{D}$, only the samples collected from users in the same cluster $\mathcal{V}(j_{u_{\text{test}}})$ are homogeneous with the test user. Therefore, the total number of homogeneous samples available for $u_{\text{test}}$ is

$$N_{\mathcal{V}(j_{u_{\text{test}}})} = \sum_{v \in \mathcal{V}(j_{u_{\text{test}}})} N_v.$$

By applying Theorem 2 from Li et al. (2022) with parameters $\Lambda = 2\sqrt{2}$, $p = q = 2$, and confidence bound term $\text{CB}_q(\Lambda) = \mathcal{I}_{u_{\text{test}}}$, we directly obtain Theorem 4.4. This completes the proof.

$$\square$$

## E  TECHNICAL LEMMAS

We introduce several lemmas which are used in our proof.

**Lemma E.1.** *Suppose $(\boldsymbol{a}_1, R), \ldots, (\boldsymbol{a}_n, r_n), \ldots$ are generated sequentially from a linear model such that $\|\boldsymbol{a}_s\|_2 \leq 1$ for all $s$, $\mathbb{E}[r_s|\boldsymbol{a}_s] = \boldsymbol{\theta}^\top \boldsymbol{a}_s$ for fixed but unknown $\boldsymbol{\theta}$, and $\{r_s - \boldsymbol{\theta}^\top \boldsymbol{a}_s\}_{s=1,2,\cdots}$ have $R$-sub-Gaussian tails. Let $M_n = \lambda I + \sum_{s=1}^n \boldsymbol{a}_s \boldsymbol{a}_s^\top$, $\boldsymbol{b}_n = \sum_{s=1}^n r_s \boldsymbol{a}_s$ and $\delta > 0$. Let $\hat{\boldsymbol{\theta}}_s = M_s^{-1} \boldsymbol{b}_s$ is the ridge regression estimator of $\boldsymbol{\theta}$. Then*

$$\left\| \hat{\boldsymbol{\theta}}_n - \boldsymbol{\theta} \right\|_{M_n} \leq R \sqrt{d \log\left(1 + \frac{n}{\lambda d}\right) + 2 \log\left(\frac{1}{\delta}\right)} + \sqrt{\lambda}$$

*holds with probability at least $1 - \delta$.*

**Lemma E.2.** *Let $\boldsymbol{a}_s$ for $s \in [n]$ be samples generated under Assumption 2.1 and $M_n = \sum_{s=1}^n \boldsymbol{a}_s \boldsymbol{a}_s^\top$. Then event*

$$\lambda_{\min}(M_n) \geq \frac{1}{2} \tilde{\lambda}_a n, \ \forall n \geq \frac{16}{\tilde{\lambda}_a^2} \log\left(\frac{8d}{\tilde{\lambda}_a^2 \delta}\right)$$

*holds with probability at least $1 - \delta$.*

**Lemma E.3.** *Let $\boldsymbol{a}_s$ for $s \in [n]$ be samples generated under Assumption B.1 and $M_n = \sum_{s=1}^n \boldsymbol{a}_s \boldsymbol{a}_s^\top$. Then event*

$$\lambda_{\min}(M_n) \geq \left( n\lambda_a - \frac{1}{3}\sqrt{18nA(\delta) + A(\delta)^2} - \frac{1}{3}A(\delta) \right)$$

*holds with probability at least $1 - \delta$ for $n \geq 0$ where $A(n, \delta) = \log\left(\frac{(n+1)(n+3)d}{\delta}\right)$. Furthermore,*

$$\lambda_{\min}(M_n) \geq 2\lambda_a n > \frac{1}{2}\lambda_a n, \ \forall n \geq \frac{16}{\lambda_a^2} \log\left(\frac{8d}{\lambda_a^2 \delta}\right)$$

*holds with probability at least $1 - \delta$.*

**Lemma E.4.** *If $a, b > 0$ and $ab > e$, then $\forall t \geq 2a \log(ab)$, we have $t \geq a \log(bt)$ holds.*

**Lemma E.5.** *When $n \geq \frac{512d}{\gamma^2 \tilde{\lambda}_a} \log(\frac{2U}{\delta})$ and $\lambda \leq d \log(1 + \frac{n}{\lambda d}) + 2 \log\left(\frac{2U}{\delta}\right)$,*

$$\frac{\sqrt{d \log(1 + \frac{n}{\lambda d}) + 2 \log(\frac{2U}{\delta})} + \sqrt{\lambda}}{\sqrt{\tilde{\lambda}_a \lambda + \tilde{\lambda}_a n/2}} \leq \frac{\gamma}{4}.$$

The proofs of the above lemmas can be found in Li & Zhang (2018) and Wang et al. (2025). Specifically, Lemma E.1 corresponds to Lemma 1, Lemma E.4 to Lemma 9, and Lemma E.5 to Lemma 10 in Li & Zhang (2018). Furthermore, Lemma E.2 comes from combining Assumption 2.1, Lemma J.1 in Wang et al. (2023a) and Lemma 7 in Li & Zhang (2018), while Lemma E.3 is from Lemma 2 in Li et al. (2025), Lemma 7 in Li & Zhang (2018) and Lemma B.2 in Wang et al. (2025).

# F ADDITIONAL EXPERIMENTS

## F.1 BASIC SETUP AND DATASET DESCRIPTION

**Dataset.** We fix the number of users at $U = 1\mathrm{k}$ and the preference vector dimension at $d = 20$. The dataset size $|\mathcal{D}|$ ranges from 5k to 100k in the insufficient-data setting and from 0.2M to 1M in the sufficient-data setting. For each experiment, the first half of $|\mathcal{D}|$ is used for offline training and the second half for evaluation. All experiments are repeated 10 times with random seeds, with parameters fixed as $\lambda = 0.5$, $\alpha = 0.1$, and $\delta = 0.01$. For ARMUL, we set the learning rate $\eta = 0.01$, the number of iterations $T_{\text{iteration}} = 10$, and performed an $n_{\text{fold}} = 2$ uniform split of the dataset.

(1) *Synthetic Dataset:* User preference vectors $\boldsymbol{\theta}_u$ and item feature vectors $\boldsymbol{a}_u^i$ lie in a $d = 20$-dimensional space, with entries drawn from a standard Gaussian distribution and normalized to unit norm. Users are uniformly partitioned into $J = 10$ clusters. For each user, the candidate action set $\mathcal{A}_u$ contains 20 items, whose feature vectors are sampled independently from a Gaussian distribution and normalized. Rewards are simulated as

$$r_u^i = \langle \boldsymbol{\theta}_u, \boldsymbol{a}_u^i \rangle + \epsilon_u^i,$$

where $\epsilon_u^i$ is zero-mean sub-Gaussian noise with variance parameter $\sigma = 0.05$.

(2) *Real-World Datasets:* We select the top 1,000 users and items by interaction frequency and construct the user–item feedback matrix. User preference vectors are extracted via Singular Value Decomposition (SVD) following Li et al. (2019), i.e.,

$$R_{U \times M} = \Theta S X^\top, \quad \Theta = (\boldsymbol{\theta}_u)_{u \in \mathcal{U}}, \quad X = (x_j)_{j \in [M]}.$$

In this setting, user preferences are scattered and exhibit clustering behavior, which differs from the synthetic assumption. Each user's candidate action set $\mathcal{A}_u$ again contains 20 items, generated by sampling sub-Gaussian feature vectors and normalizing them to unit norm.

**User Probability Distribution.** To allocate the dataset size across users, we consider two sampling strategies:

- **Equal Distribution:** Each user is sampled with equal probability, i.e.,

$$\mathbb{E}[|\mathcal{D}_u|] = \frac{|\mathcal{D}|}{U}.$$

- **Semi-Random Distribution:** Sampling is uniform within each cluster, but probabilities differ across clusters. Let $j_u$ denote the cluster index of user $u$, and $\mathcal{V}(j_u)$ the set of users in cluster $j_u$. Then

$$\mathbb{E}[|\mathcal{D}_u|] = \frac{|\mathcal{D}|}{|\mathcal{V}(j_u)|} \cdot p_{j_u}, \quad p = [p_1, \ldots, p_J], \quad \sum_{j=1}^J p_j = 1, \quad p_j \geq 0.$$

## F.2 BENCHMARK METHODS AND THEIR DESCRIPTIONS

To ensure fairness, we simulate online algorithms (e.g., CLUB) in the offline setting by treating the dataset as a one-shot interaction stream. Each algorithm runs its original policy to make predictions without further exploration. Clustering is done once before the evaluation stage, avoiding dynamic updates and preserving the offline nature. It is important to note that, without modification, the original method may lead to meaningless updates during offline training. Specifically, in the case of CLUB, removing edges under insufficient information can cause irreversible loss of structural information.

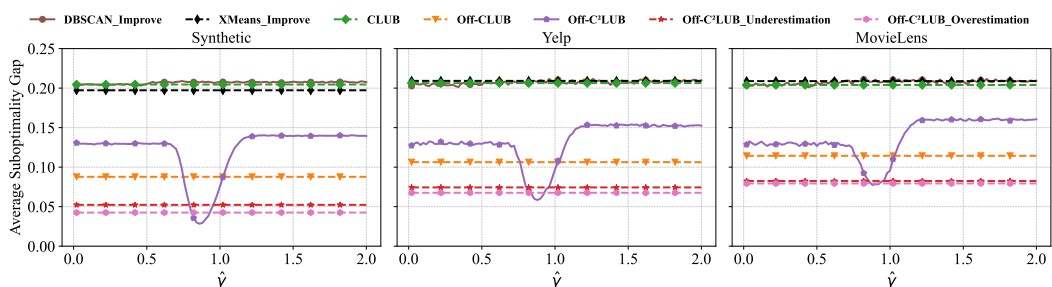

Figure 4: Comparison of clustering strategies under $|\mathcal{D}| = 30k$ across three datasets.

**Baseline Algorithms.** To ensure comprehensive and fair comparisons, we select the following baseline algorithms from existing studies. We adapt several online learning algorithms to the offline setting, modifying them as necessary to work effectively with offline datasets. Additionally, we propose enhanced versions of DBSCAN (Schubert et al., 2017), specifically tailored to our experimental environment. For these multi-armed bandit (MAB) methods, we first use offline training data to update model parameters, such as the preference vector $\boldsymbol{\theta}$. We then refine the graph or set structure based on the updated parameters.

For offline clustering algorithms. We incorporate DBSCAN_Improve and XMeans_Improve to replace the clustering mechanism in CLUB. Furthermore, we uniformly augment all compared methods by introducing an upper confidence bound (UCB) term to ensure fair comparison. The evaluated baseline methods include:

- **LinUCB_Ind** (Abbasi-Yadkori et al., 2011): An independent variant of LinUCB that operates without leveraging any clustering information. This method applies the LinUCB approach independently to each user.

- **CLUB** (Gentile et al., 2014): A cluster-based multi-armed bandit method that groups users from a complete graph according to their characteristics to improve model selection.

- **DBSCAN_Improve** (Schubert et al., 2017): A variant of CLUB that replaces its original clustering mechanism with the DBSCAN algorithm.

- **XMeans_Improve** (Pelleg et al., 2000): A variant of CLUB that replaces its original clustering mechanism with the X-Means algorithm.

- **SCLUB** (Li et al., 2019): A method that employs a hierarchical clustering structure to model users and enhances learning efficiency by optimizing cluster centers.

- **Off-CLUB** (Algorithm 2): A cluster-based multi-armed bandit method that groups users from a complete graph according to their characteristics to improve model selection. It utilizes a Lower Confidence Bound (LCB) exploration strategy and performs aggregation through neighbor information.

- **Off-C²LUB** (Algorithm 1): A cluster-based multi-armed bandit method that groups users from a null graph based on their characteristics to enhance model selection. It employs an LCB exploration term and aggregates feedback via neighboring nodes. The $\hat{\gamma}$ is selected to be approximate to the optimal value.

- **Off-C²LUB_Underestimation**: A variant of Off-C²LUB that uses an underestimation-based estimation policy to compute $\hat{\gamma}$.

- **Off-C²LUB_Overestimation**: A variant of Off-C²LUB that employs an overestimation-based estimation policy to determine $\hat{\gamma}$.

- **ARMUL** (Duan & Wang, 2023): A framework named Adaptive and Robust MUlti-task Learning (ARMUL) addresses the multi-task learning problem. Use a regularization term to balance the similarity and differences between tasks.

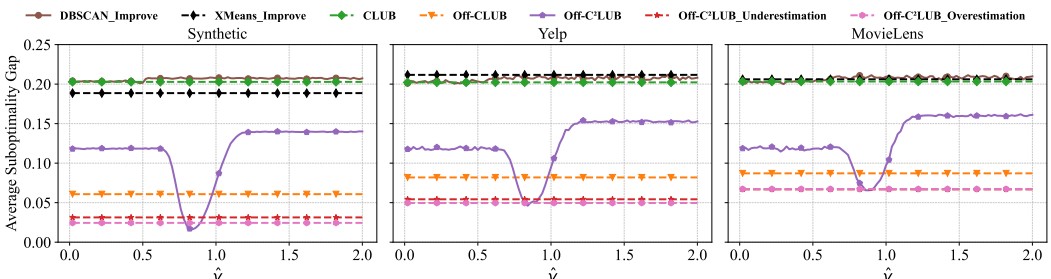

Figure 5: Comparison of clustering strategies under $|\mathcal{D}| = 35k$ across three datasets.

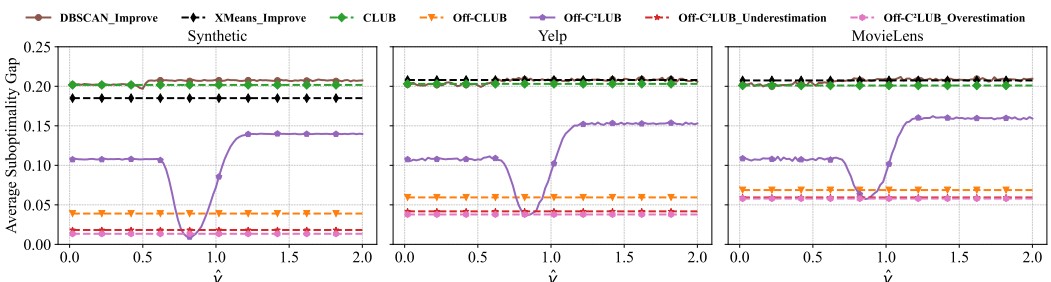

Figure 6: Comparison of clustering strategies under $|\mathcal{D}| = 40k$ across three datasets.

### F.3 OFFLINE DATA DISTRIBUTION (DETAILED ANALYSIS)

The offline dataset is influenced by the user's strategy during the sampling phase. For each data point in the dataset $\mathcal{D}_u$, a subset of 20 items ($\mathcal{A}_u$) is generated for sampling. We consider two selection strategies:

- *Random Selection*: The user randomly selects one item from $\mathcal{A}_u$, denoted as $\boldsymbol{a}_u^i \sim \text{Uniform}(\mathcal{A}_u)$. Figure 2 shows the experimental results.

- *LinUCB Selection*: During the offline phase, each user independently applies the LinUCB method (Abbasi-Yadkori et al., 2011) to select one item from $\mathcal{A}_u$, denoted as $\boldsymbol{a}_u^i = \underset{\boldsymbol{a} \in \mathcal{A}_u}{\operatorname{argmax}} \langle \boldsymbol{\theta}_u, \boldsymbol{a} \rangle + \alpha \sqrt{\boldsymbol{a}^\top M_u^{-1} \boldsymbol{a}}$. Figure 7 shows the experimental results.

When data is collected randomly, it suggests that users are choosing items without demonstrating explicit preferences. In this case, the collected data covers all feature dimensions in a relatively uniform manner, ensuring broad coverage but lacking targeted exploration. In contrast, a more informed collection strategy, such as LinUCB, enables the system to identify and focus on more informative dimensions. Although this process still relies on item–reward pairs from different arms, it guides the exploration of the feature space in a more deliberate way. As a result, the algorithm achieves tighter confidence intervals and improved estimation accuracy.

In summary, while random selection guarantees comprehensive coverage of the feature space, Lin-UCB offers more targeted exploration, ultimately enhancing algorithmic performance. From a practical perspective, if a recommendation system has already been operating under LinUCB, it can seamlessly transition to the Off-C²LUB algorithm while reusing previously collected data. Conversely, if a newly designed algorithm requires uniformly distributed data, it may be necessary to re-run the system to gather data consistent with such assumptions.

### F.4 PERFORMANCE ANALYSIS

**Performance with Insufficient Dataset.** To illustrate the performance under insufficient data with LinUCB selection, we report the average suboptimality gap across user distributions and dataset

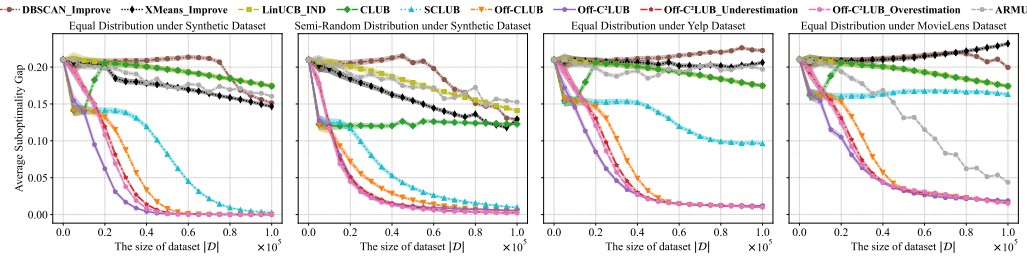

Figure 7: Suboptimality Gap under Different User Probability Distributions and LinUCB Sampling Strategy with Insufficient Data.

Table 6: Impact of Increased Data on Suboptimality gap under Yelp Dataset.

| Algorithm \ Dataset size | 0.2M | 0.4M | 0.6M | 0.8M | 1M |
|---|---|---|---|---|---|
| DBSCAN_Improve | 0.146616 | 0.069673 | 0.033521 | 0.018761 | 0.011276 |
| XMeans_Improve | 0.178563 | 0.138018 | 0.103209 | 0.089212 | 0.086906 |
| CLUB | 0.144821 | 0.066241 | 0.028270 | 0.013642 | 0.007569 |
| Off-CLUB | 0.007643 | 0.002359 | 0.001360 | 0.000968 | **0.000751** |
| LinUCB_IND | 0.144928 | 0.066318 | 0.028251 | 0.013649 | 0.007584 |
| SCLUB | 0.097121 | 0.045636 | 0.020032 | 0.010014 | 0.005680 |
| ARMUL | 0.137016 | 0.112671 | 0.095382 | 0.085142 | 0.070668 |
| Off-C$^2$LUB | 0.010966 | 0.009098 | 0.008366 | 0.003407 | 0.000778 |
| Off-C$^2$LUB_Underestimation | 0.007720 | 0.002684 | 0.001758 | 0.001340 | 0.001126 |
| Off-C$^2$LUB_Overestimation | **0.007579** | **0.002353** | **0.001345** | **0.000950** | 0.000751 |

Table 7: Impact of Increased Data on Suboptimality gap under MovieLens Dataset.

| Algorithm \ Dataset size | 0.2M | 0.4M | 0.6M | 0.8M | 1M |
|---|---|---|---|---|---|
| DBSCAN_Improve | 0.150422 | 0.071688 | 0.031250 | 0.016067 | 0.009879 |
| XMeans_Improve | 0.202409 | 0.190408 | 0.178008 | 0.168371 | 0.160795 |
| CLUB | 0.144998 | 0.066608 | 0.028231 | 0.013636 | 0.007575 |
| Off-CLUB | 0.010777 | 0.002992 | **0.001580** | **0.001077** | **0.000818** |
| LinUCB_IND | 0.145213 | 0.066277 | 0.028228 | 0.013666 | 0.007573 |
| SCLUB | 0.140205 | 0.064720 | 0.027545 | 0.013374 | 0.007443 |
| ARMUL | 0.119995 | 0.089696 | 0.091467 | 0.078193 | 0.057992 |
| Off-C$^2$LUB | 0.016508 | 0.014446 | 0.013647 | 0.013440 | 0.013294 |
| Off-C$^2$LUB_Underestimation | 0.011465 | 0.004185 | 0.002810 | 0.002353 | 0.002099 |
| Off-C$^2$LUB_Overestimation | **0.010776** | **0.002980** | 0.001589 | 0.001081 | 0.000821 |

sizes (Figure 7). All results are averaged over 10 random runs, and the error bars are computed using standard errors. For Off-C$^2$LUB, $\hat{\gamma}$ is optimized under the equal distribution scenario and applied to the semi-random distribution. With $|\mathcal{D}| \in [20k, 100k]$, both Off-C$^2$LUB variants significantly reduce the suboptimality gap. On the synthetic dataset, Off-C$^2$LUB_Overestimation improves over Off-CLUB by 45.4% and over other baselines by at least 76.3%, while Off-C$^2$LUB_Underestimation achieves 33.7% and 71.3%, respectively. On Yelp, the corresponding improvements are 27.2% and 76.5% for Overestimation, and 21.4% and 74.6% for Underestimation. On MovieLens, they are 17.2% and 75.3% for Overestimation, and 14.4% and 63.4% for Underestimation. Overall, Off-C$^2$LUB achieves the best performance, while Off-CLUB, though less effective, still outperforms classical clustering baselines.

**Performance with Sufficient Data.** Table 6 and Table 7 further report the convergence performance in the real dataset under sufficient data. The bold values mark the optimal suboptimal gaps (rounded to six decimals) for each dataset size. Our proposed Off-CLUB and Off-C$^2$LUB consistently outperform all other baselines. Compared to classical methods such as CLUB and SCLUB, our approaches converge faster and consistently maintain a much smaller suboptimality gap, underscoring their efficiency and robustness.

**Improvement with Estimated $\hat{\gamma}$.** We evaluate different strategies under limited data conditions using the synthetic, Yelp, and MovieLens datasets with $|\mathcal{D}| = 30k$ (Figure 4), $|\mathcal{D}| = 35k$ (Figure 5), and $|\mathcal{D}| = 40k$ (Figure 6). Solid lines denote methods using $\hat{\gamma}$, and dashed lines those without. Consistent with the main text, our algorithm achieves near-optimal performance compared to the Off-C$^2$LUB lower bound. This result holds across different dataset scales and domains, regardless of the $\hat{\gamma}$ estimation method, demonstrating that both estimation strategies maintain stable and competitive performance.

## G FURTHER DISCUSSIONS ON OFF-CLUB

### G.1 DETAILED ALGORITHM: OFF-CLUB

The pseudo-code of Off-C$^2$LUB is shown in Algorithm 2. The algorithm takes similar inputs to Algorithm 1, except that it does not require $\hat{\gamma}$. It starts with a complete graph $\mathcal{G}$ and iterates through all edges $(u_1, u_2) \in \mathcal{E}$ during the Cluster Phase. If the condition $\left\| \hat{\boldsymbol{\theta}}_{u_1} - \hat{\boldsymbol{\theta}}_{u_2} \right\|_2 > \alpha \left( \mathrm{CI}_{u_1} + \mathrm{CI}_{u_2} \right)$ holds, the algorithm concludes that users $u_1$ and $u_2$ do not share the same preference vector and

removes the edge between them (Line 5). After processing all edges, the resulting graph $\tilde{\mathcal{G}}$ separates users with different preference vectors into distinct clusters, provided that each user has sufficient samples to ensure a low confidence interval. However, if a user $u$ has fewer samples in $\mathcal{D}_u$, the resulting higher confidence interval $\text{CI}_u$ reduces the likelihood of satisfying condition (6), potentially leading to the inclusion of heterogeneous users in the same cluster. The conditions necessary to ensure the algorithm's correctness are discussed in detail in Section 4.2.

---

**Algorithm 2** Off-CLUB

---

1: **Input:** User $u_{\text{test}}$, action set $\mathcal{A}_{\text{test}}$, dataset $\mathcal{D}$, parameters $\alpha$, $\lambda > 1$ and $\delta > 0$
2: **Initialization:** Construct a complete graph $\mathcal{G} = (\mathcal{V}, \mathcal{E})$ where $\mathcal{V} = \mathcal{U}$ is the set of all users, and compute $M_u$, $\boldsymbol{b}_u$, $\hat{\boldsymbol{\theta}}_u$, and $\text{CI}_u$ as in (1) for each user $u \in \mathcal{V}$
3: \\Cluster Phase
4: **for** each edge $(u_1, u_2) \in \mathcal{E}$ **do**
5:     Remove edge $(u_1, u_2)$ if condition (6) is satisfied
6: **end for**
7: Let $\tilde{\mathcal{G}} = (\mathcal{V}, \tilde{\mathcal{E}})$ denote the updated graph
8: **for** each user $u \in \mathcal{V}$ **do**
9:     Aggregate data:

$$\tilde{\mathcal{D}}_u = \bigcup_{(u,v) \in \tilde{\mathcal{E}}} \left\{ (\boldsymbol{a}_v^i, r_v^i) \in \mathcal{D}_v \right\} \cup \mathcal{D}_u,$$

10:     Compute cluster statistics:

$$\tilde{M}_u = \lambda I + \sum_{(\boldsymbol{a},r) \in \tilde{\mathcal{D}}_u} \boldsymbol{a}\boldsymbol{a}^\top, \quad \tilde{N}_u = \left| \tilde{\mathcal{D}}_u \right|, \quad \tilde{\boldsymbol{b}}_u = \sum_{(\boldsymbol{a},r) \in \tilde{\mathcal{D}}_u} r\boldsymbol{a}, \quad \tilde{\boldsymbol{\theta}}_u = \left( \tilde{M}_u \right)^{-1} \tilde{\boldsymbol{b}}_u$$

11: **end for**
12: \\Decision Phase
13: Select the action:

$$\boldsymbol{a}_{\text{test}} = \underset{\boldsymbol{a} \in \mathcal{A}_{\text{test}}}{\arg\max} \left( \left( \tilde{\boldsymbol{\theta}}_{u_{\text{test}}} \right)^\top \boldsymbol{a} - \beta \, \|\boldsymbol{a}\|_{\left( \tilde{M}_{u_{\text{test}}} \right)^{-1}} \right)$$

where $\beta = \sqrt{d \log \left( 1 + \frac{\tilde{N}_{u_{\text{test}}}}{\lambda d} \right) + 2 \log \left( \frac{2U}{\delta} \right)} + \sqrt{\lambda}$

---

Similar to Algorithm 1, this algorithm aggregates data from each user and its **neighbors** in $\tilde{\mathcal{G}}$ to form a new dataset and computes the necessary statistics (Lines 9 and 10). It is important to note that this data aggregation strategy, like that in Algorithm 1, differs from the online version of the CLUB algorithm introduced in (Gentile et al., 2014), where data from all users within the same connected component of the graph is utilized. For instance, consider a scenario where $(u,v) \in \tilde{\mathcal{E}}$ and $(v,w) \in \tilde{\mathcal{E}}$, but $(u,w) \notin \tilde{\mathcal{E}}$. In this case, samples from $\mathcal{D}_w$ are excluded from $\tilde{\mathcal{D}}_u$ in Algorithm 2, whereas the online version includes them in $\tilde{\mathcal{D}}_u$. The restricted aggregation in Algorithm 2 is crucial in the offline setting: since users that are no longer neighbors of $u$ are guaranteed not to share the same preference vector, including their data in $\tilde{\mathcal{D}}_u$ would introduce bias. In the online setting, however, such bias is not a concern because infinite online data ensures that all edges between users with different preference vectors are reliably removed. This distinction highlights the importance of careful data selection in the offline setting to maintain prediction accuracy.

Finally, the Decision Phase is performed in a manner analogous to Algorithm 2, using a pessimistic estimate to select $\boldsymbol{a}_{\text{test}}$. This pessimistic strategy differs from the optimistic approaches used in prior online clustering of bandits (Gentile et al., 2014; Li & Zhang, 2018; Li et al., 2019; Wang et al., 2025; Li et al., 2025), which balance the exploration–exploitation tradeoff. Instead, it follows the pessimism principle widely adopted in offline bandits and RL (Jin et al., 2021; Li et al., 2022; Liu et al., 2025), avoiding over-reliance on poorly represented dimensions.

## G.2 ADDITIONAL REMARKS

*Remark* G.1 (Asymptotic Properties). In the special case where $\mathcal{V}(j_{u_{\text{test}}}) = \{u_{\text{test}}\}$ (i.e., the test user is distinct from all other users in $\mathcal{V}$), the result reduces to the standard bound for linear bandits, $\tilde{O}\left(\sqrt{\frac{d}{\tilde{\lambda}_a N_{u_{\text{test}}}}}\right)$. Conversely, when $\mathcal{V}(j_{u_{\text{test}}}) \approx \mathcal{V}$ (i.e., most users share the same preference vector), Algorithm 2 effectively aggregates data from nearly all users, leveraging the large sample size to make more accurate predictions.

*Remark* G.2 (Intuition of Assumption 4.1). The key assumption underlying Theorem 4.2 is data sufficiency. As defined in Assumption 4.1, complete data sufficiency comprises two main components. The first component depends on $1/\tilde{\lambda}_a^2$, which ensures sufficient information about each dimension of the preference vector for each user. Without this level of data, the algorithm cannot reliably estimate each dimension, making accurate predictions infeasible. The second component in Assumption 4.1 depends on $1/\gamma^2$, which is the main part of the data sufficiency requirement. This term guarantees that the true cluster of each user can be accurately identified. Insufficient data in this regard prevents the algorithm from correctly assigning users to their respective clusters, potentially leading to the inclusion of users with different preference vectors and introducing bias into the predictions. Notably, a smaller $\gamma$ corresponds to a stricter data sufficiency requirement.

