# OpenReview forum: "Offline Clustering of Linear Bandits: The Power of Clusters under Limited Data"
_ICLR.cc/2026/Conference — Submitted to ICLR 2026_

### Official Review · Reviewer_vvYj · 2025-10-20

**Soundness:** 3
**Presentation:** 4
**Contribution:** 3
**Rating:** 6
**Confidence:** 3

**Summary:**

The paper considers making an item recommendation with offline data where users belong to clusters (users in the same cluster share the same preference vector).   A test user has already made some ratings.  These ratings are used to select a cluster of similar users, the preference vector of this cluster is estimated and used to select an item to recommend to the test user.  It appears that the same item can be recommended multiple times to the same user, although the text is not completely clear on this.

**Strengths:**

The paper is well written, the analysis seems sound although I haven't checked all of the maths.

**Weaknesses:**

The proposed algorithm seems similar to a nearest neigbours approach, which have been very well studied in recommendation systems.  I would like to see more discussion on this since it is directly relevant to the contribution.  I think I understand why the paper focuses solely on offline data, but a more useful setup would be where offline data is used to pre-train a recommender system which then carries out online learning (e.g. this is a v common setup in the recommender RL literature).

The paper is not the first work on clustered recommendation.  See for example "Cluster-Based Bandits: Fast Cold-Start for Recommender System New Users", https://dl.acm.org/doi/abs/10.1145/3404835.3463033, "Fast and Accurate User Cold-Start Learning Using Monte Carlo Tree Search" https://dl.acm.org/doi/abs/10.1145/3523227.3546786.

Regarding the theory contribution, the regret bound is a probabilistic worst-case one.  Its not clear that worst-case bounds are especially interesting or useful and the paper would be greatly strengthened by presenting more evidence on whether the worst-case analysis is much different from the observed regret.

**Questions:**

See comments in weaknesses section above

---

> ### Author Response · Authors · 2025-11-21
>
> We sincerely thank the reviewer for the valuable comments and high recognition of our work. We address the reviewer's concerns as follows.
>
> **W1: Comparisons of nearest neighbor search methods.** While also focus on finding users with similar preferences and widely adopted in recommendation systems, our approaches in offline bandits with traditional nearest neighbor search methods (such as KNN) have many fundamental differences. Firstly, which is the most important one, our setting of offline clustering of bandits is in face of the limited data challenge, which drives us to design algorithms that incorporate the number of data and confidence of estimations for each user. Traditional nearest neighbor search methods mainly utilize the estimated parameters to select those with similar preferences. However, in our setting, a crucial challenge is that limited data per user will make the estimations noisy and thus not reliable if the data is insufficient. Hence, our algorithm utilizes the input cluster threshold $\hat\gamma$ to estimate and control the gap between users with similar preferences, and designs the confidence interval for each user to account for the confidence of estimation accuracy, which depends on the number of samples used. Using this protocol, the cluster condition in Eq.(2) of Section 3.1 helps control the bias between connected users and is able to tackle the scenarios with limited data. This is also verified by our theoretical results in Lemma 3.2 and Theorem 3.4 where the bias for heterogeneous users is bounded by $\hat\gamma$. By contrast, traditional nearest neighbor algorithms do not account for the influence of limited data, which restricts their availability in our setting. Another crucial difference is that our algorithm goes beyond simply learning the cluster and also focuses on high-quality decision-making. Specifically, in the decision phase our algorithm utilizes aggregated data based on the learned cluster property, adopts the pessimism estimates based on the aggregated information matrix and selects actions accordingly, the algorithm quality is also determined by the suboptimality caused by action selection rather than the learned cluster. This reflects the difference between both settings, where our setting needs not only to learn the cluster property accurately, but also to make high quality decisions.
>
> Regarding the reviewer's suggestion on pre-training a recommender system based on the offline data, we can give some intuitions here. One potential solution is to use the offline data to pre-learn the cluster property, just as in the cluster phase in our first algorithm. The algorithm can rely on the learned cluster to aggregate data and make decisions for the coming online requests when switching to the online phase. Moreover, the algorithm can also learn the cluster property better with these online data. Throughout this way, the algorithm is able to utilize the offline data and make more reliable decisions with the pre-trained model, while keeps updating parameters with the coming online data. While our focus on is the pure offline model for clustering of bandits problem, we leave this potential setting in recommender systems with pre-trained offline model with online learning adaptations as an invaluable future work.
>
> **W2: Related works in clustered recommendation.** Thanks for pointing out. These papers are mainly in recommender settings where users have different ratings for different items. Many parts in the setting including the rewards, feedback, assumptions, learning goal etc. are largely different with our setting, while sharing some similarities in also having some cluster properties. We would be happy to add them in our discussions and comparisons of related works in the future versions.

---

> > ### Author Response · Authors · 2025-11-21
> >
> > **W3: Difference between worst-case regret and observed regret.**
> > We clarify that the theoretical bounds in Sections 3 and 4 provide probabilistic worst-case upper bounds on the true suboptimality (i.e., one-shot decision regret) as defined in Section 2, which is the standard performance measure in offline linear bandits literature. These bounds characterize how our algorithms behave in the worst case, while the observed suboptimality is evaluated empirically in Section 5. Importantly, our empirical results strongly confirm that the worst-case analysis is well aligned with the observed behavior. In Figure 2, the purple, red, pink, and orange curves show that the empirical suboptimality of our methods closely follows the theoretical rate: the suboptimality decreases proportionally to $1/\sqrt{N}$, where $N$ is the number of offline samples used. Moreover, Figure 3 further verifies our theoretical noise-bias tradeoff: both overly small and overly large values of $\hat{\gamma}$ degrade performance, exactly as predicted by our analysis in Lemma 3.2 and Theorem 3.4. Overall, the experiments demonstrate the accuracy of our worst-case analysis. It accurately captures the key effects of limited offline data and heterogeneous neighbor bias that drive performance in the offline clustering of bandits setting.

---

### Official Review · Reviewer_dbFf · 2025-10-30

**Soundness:** 3
**Presentation:** 2
**Contribution:** 2
**Rating:** 2
**Confidence:** 4

**Summary:**

This paper addresses the problem of pooling offline data from similar users to a current user whom the learner has to make decisions for. Two algorithms are provided. The first amounts to estimating a minimum cluster distance, constructing a graph where two users are connected if their estimated parameters are closer than the minimum distance minus a confidence bound penalty, and then pooling data for users adjacent to the test user together when making decisions for the test user. The second is deferred to the appendix. Numerical experiments show outperformance relative to online algorithms, but not offline ones.

**Strengths:**

The authors consider a salient problem of pooling data from heterogenous users together to make decisions in adapting to a new user. The paper is largely rigorously written, and the proofs appear to be sound. The experiments are involved, especially for a theory paper.

**Weaknesses:**

1. The authors are very much unaware of a large body of existing literature in this field.
- The "clusters of bandits" problem is essentially an offline version of the latent bandit problem. See Hong et al. (2020), who first learn clusters from offline data before taking actions online. Shi et al. (2023) discuss offline latent RL, which is a variant of the latent MDP setting of Kwon et al. (2021).
- Rigorous guarantees for clustering tabular MDPs under the Markovian setting have been obtained by Kausik et al. (2023), with application to learning a policy from an offline dataset of clustered users in Kausik et al. (2024). The same authors deal with offline data in the linear latent contextual bandit setting in Kausik et al. (2025), adapting to a test user given data from other users as in this paper, but for online exploration.
- Clusters of linear bandit instances, or the more broad multi-task setting, have also been addressed in the following:
    - Yang et al. (2021) discuss a setting where the learner can play $k$ linear bandits in dimension $d$ concurrently, with a shared representation.
    - Hu et al. (2021) solve the aforementioned problem, except with infinite actions, with a projected LinUCB-like algorithm, and extend it to reinforcement learning.
    - Yang et al. (2022) close a gap that Yang et al. (2021) leave in the infinite action setting.
These are highly relevant, and should be included in the paper. This is a non-exhaustive list. It is highly alarming that the authors do not seem to be aware of this body of work. This renders some of their claims of novelty suspect.
2. Accordingly, this influences the guarantees that the authors provide.
- The algorithm is simple and barely involves any clustering. It amounts to connecting any two user if the estimated parameters are closer than the estimated minimum distance minus a confidence bound, and there is sufficient data for each user. Data from users connected to the test user is pooled together to form a pessimistic estimate of the test user's reward parameter, from which an action is chosen.
- No data from non-neighbors of the test user is used, which renders the computation for those vertices moot. The algorithm would have been better presented as "Select all users close enough to the test user, then pool their data to form a pessimistic estimate" -- though this looks less impressive. This can be done lazily per test user, with the result cached for future use. The precomputation is not necessary.
- Algorithm 1 peculiarly requires an estimate of the minimum gap between clusters. This is never known in practice, so it is odd that the authors consider the case where it is. The guarantees for the underestimation and overestimation policies are rather weak, and only discussed in the appendix. Other work (e.g. Kausik et al. (2023)) estimates a similar quantity from data and provides an end-to-end guarantee.
    - Given the minimum gap, the problem is then very simple! It reduces to finding all users that are within the minimum gap minus a confidence-based penalty to be conservative -- which is exactly what Algorithm 1 is. There is no clustering that needs to be done whatsoever.
    - A few other quantities (e.g. the smoothed regularity parameter) are also assumed to be known, which is quite a strong assumption.
    - Other assumptions include full coverage over the action space (i.e. a full-rank covariance matrix with bounded minimum eigenvalue), and actions within the offline data that are independently drawn from some distribution. I believe that it is folklore that these may be necessary, but why exactly this is necessary is never discussed beyond appealing to prior work.
- Algorithm 2 is named Off-CLUB but addressed as Off-C^2LUB on Line 1508.


### References
- Hong et al. (2020), Latent bandits revisited
- Shi et al. (2023), Provably Efficient Offline Reinforcement Learning with Perturbed Data Sources
- Kwon et al. (2021), RL for latent mdps: Regret guarantees and a lower bound.
- Kausik et al. (2023), Learning mixtures of Markov chains and MDPs
- Kausik et al. (2024), Offline Policy Evaluation and Optimization under Confounding
- Kausik et al. (2025), Leveraging Offline Data in Linear Latent Contextual Bandits
- Yang et al. (2021), Impact of representation learning in linear bandits
- Hu et al. (2021) Near-optimal representation learning for linear bandits and linear RL
- Yang et al. (2022) Nearly minimax algorithms for linear bandits with shared representation

**Questions:**

1. How does your paper relate to the existing literature?
2. Can Algorithm 1 be simplified as above to nearest neighbor search?
3. Can you provide any offline bandit baselines?

This score is somewhat harsh. At the moment, I am on the fence between a 2 and a 4, and am willing to increase my score if I am proven wrong or my concerns are addressed.

---

> ### Author Response · Authors · 2025-11-21
>
> Thank you for your valuable comments. We summarize your concerns in the following five perspectives and reply accordingly.
>
> **(i) Comparison of literature.** We thank the reviewer for highlighting these relevant works. While “latent bandits’’ and “multi-task learning’’ share the general idea of exploiting structure, the **problem settings**, **challenges**, and **structural assumptions** differ fundamentally from ours, leading to **distinct algorithms and theoretical results**:
>
>
> *Comparison with latent bandit works:* The key distinction is *pure offline learning vs. online/hybrid exploration*. Prior latent-structure bandit works [5, 8, 9, 10] operate in online or hybrid environments where the learner actively pulls arms and obtains sufficient data (typically requiring dataset size $N=\Omega(T)$) to achieve sublinear regret. Their objective is cumulative regret minimization, and errors made in early clustering can be corrected through future interactions. In contrast, our pure offline setting addresses a strictly offline and data-limited regime, often with fixed, limited and even sparse samples per user, and evaluates the suboptimality gap of a one-shot decision. Because no additional data can be collected, clustering errors are unavoidable, and the problem is governed by the limited data challenge which results in a bias-variance tradeoff rather than exploration-exploitation. This necessitates a more conservative algorithmic design, including the connecting-edge policy, bottom-up merging with $N_{\min}$ checks, and one-hop aggregation, to prevent including unrestricted bias, fundamentally differing from optimistic (UCB-style) approaches used in online or hybrid works.
>
> Specifically, we compare our setting with the most state-of-the-art latent bandits work [10], which also considers offline data from different users. As emphasized in our work, the primary challenge we address is the **limited data** issue: the offline data available for each user is fixed, limited, and potentially highly imbalanced. This challenge makes standard online clustering protocols fail. Consequently, our method incorporates several mechanisms to identify and distinguish data sufficiency across users, such as the connecting-edge policy and the minimum data requirement $N_{\min}$ (Eq. (2)), and only make confident decisions based on those with sufficient data. Our theoretical results in Lemma 3.2 and Theorem 3.4 further interpret this challenge through the noise-bias tradeoff.
>
> In comparison, Theorem 2 in [10] bounds regret by the minimum of two terms: the first is a trivial linear bandit bound using only online data, while the second is obtained from leveraging offline data to learn the projection map $U_*$. However, this improvement only holds when the offline data trajectory number $N$ is sufficiently large; otherwise, the second term suffers linear regret in $T$ and collapses to the first (trivial) term. Moreover, their analysis (e.g., Appendix C, p. 28) explicitly requires $N$ to be large to justify the derivations. Thus, whether their structural learning procedure (Algorithm 1 and 2 in [10]) improves online regret **critically depends on offline sample sufficiency and only works well under sufficient data**. In contrast, our work imposes no such requirement and is explicitly designed for settings with **arbitrary and limited offline data**. Furthermore, for the sufficient data case, we also propose Algorithm 2 (Off-CLUB), which does not require knowledge of $\gamma$ and achieves a near-optimal bound in term of utilized samples (or can be implemented via Algorithm 1 with an approximate $\gamma$), showing the optimality of our protocol if offline data is guaranteed to be sufficient. This demonstrates that our contributions cover both data-sufficient and data-insufficient regimes.
>
> Moreover, we also note several key modeling differences: First, [10] assumes the coverage condition on independently generated offline data, which does not hold for many real-world scenarios (such as with online policy logged datasets). By contrast, our Assumption 2.1 only requires regularity on the candidate sets, while the chosen actions may be arbitrary and can be inter-dependent. Second, [10] requires balanced offline trajectories (all users have trajectories of equal length $H$), which is a strong and unrealistic requirement, whereas our setting allows fully imbalanced heterogeneous data availability.
>
> Therefore, while the latent bandit problem is related to our clustering of bandits setting, the **core challenge**, **problem formulation**, **techniques**, and **analysis tools** differ substantially. The two works are related but not overlapping.

---

> > ### Author Response · Authors · 2025-11-21
> >
> > *Comparison with multi-task learning / representation-learning works:*
> > Multi-task learning approaches (e.g., [11-13]) assume that task parameters lie in a shared continuous low-rank subspace, such as $\theta_u = BW_u$ for some $k$-dimensional feature extractor $B$. Their goal is to learn this subspace, reducing the effective dimension from $d$ to $k$, and they operate in online or data-rich settings where sufficient samples can be acquired for representation learning. Their performance metric is regret, not offline suboptimality. Thus, their problem formulation and guarantees do not cover our offline discrete-cluster setting. Even in online environments, the theoretical regimes differ: state-of-the-art representation-learning bandits (e.g., [13]) achieve $\tilde{O}(d\sqrt{kMT} + kM\sqrt{T})$ regret, whereas online clustering bandits achieve $\tilde{O}(d\sqrt{kMT})$, showing that directly applying representation learning can lead to suboptimal performance when discrete clustering structure is present. This highlights that the two lines of work address fundamentally different structural assumptions and therefore yield different theoretical conclusions.
> >
> >
> > *Stateless Bandits vs. Stateful MDPs:* Works such as [6, 7] focus on Markov Decision Processes (MDPs). The primary difficulty in their setting involves handling distributional shifts in state transition dynamics ($P(s'|s,a)$) and long-term credit assignment. Our work focuses on Contextual Bandits (stateless). By abstracting away transition dynamics, we provide tighter, instance-dependent bounds specifically analyzing the tradeoff between aggregation bias (from heterogeneous users) and noise reduction (from pooling data) in linear reward estimation.
> >
> >
> >
> > **(ii) Algorithm protocol.** We summarize the reviewer’s concerns in three parts: the novelty and contribution of our algorithm, why we aggregate only over one-hop neighbors rather than the full connected component, and whether it is possible to learn neighbors only for the test user. For the first point regarding algorithmic contribution, we refer the reviewer to W1 for reviewer R7Pr and W1 for reviewer 1WaV, where we provide detailed explanations. We address the remaining two questions below.
> >
> > The reason we aggregate only over one-hop neighbors rather than the entire connected component is to better control heterogeneous bias and is indeed a crucial advantage and contribution for the algorithm. When we restrict aggregation to direct neighbors, the bias introduced by heterogeneous users in $\mathcal{W}\_{\hat\gamma}(u)$ can be bounded by $\hat\gamma$, as shown in Lemma 3.2. However, if we aggregate over all users in the connected component, this guarantee breaks down. For example, consider three users $u, v, w$ where edges $(u,v)$ and $(v,w)$ exist in $\mathcal{G}\_{\hat\gamma}$ but $(u,w)$ does not. While both $\|\theta\_u - \theta\_v\|_2$ and $\|\theta\_v - \theta\_w\|_2$ can be upper bounded by $\hat\gamma$ (Lemma 3.2), no such guarantee holds for $\|\theta\_u - \theta\_w\|_2$, which may be as large as $2\hat\gamma$. This simple example shows that aggregating across the whole connected component would therefore introduce uncontrolled bias and invalidate the clean bias bound in our current analysis in Lemma 3.2 and Theorem 3.4. This rationale is also discussed in Appendix G when describing the aggregation rule for Off-CLUB, and is consistent with several prior works (e.g., Section IV.A in [1], Section 3.4 in [2]).
> >
> > Regarding the third question, we agree that if there is a single fixed test user and only a one-shot decision, one could indeed learn neighbors only for that user, reducing the complexity from $O(U^2)$ to $O(U)$. However, we intentionally learn the full cluster structure for all users to make the algorithm robust to more general and realistic scenarios: such as when test users arrive from a distribution and is not fixed or when the learned cluster structure must be reused for different users. Both settings commonly arise in practical applications. Thus, although our problem statement considers a fixed test user for clarity, our full graph learning design ensures that the algorithm naturally extends to more complex multi-test, repeated-decision environments, at the cost of a few additional computation. We will clarify this design choice more explicitly in our next revision.

---

> > > ### Author Response · Authors · 2025-11-21
> > >
> > > **(iii) Estimation of minimum gap.** Our paper provides two cases for selecting $\hat\gamma$. For the first case, where $\gamma$ is known, we note that this parameter is also assumed to be known in some prior works on online clustering of bandits [3, 4]. Our result in Corollary 3.7 shows that with known $\gamma$, the suboptimality bound becomes bias-free and depends only on noise, and under data sufficient scenarios it nearly matches the lower bound (discussed in Remark 4.5) in term of number of utilized samples, offering strong theoretical guarantees. We also highlight that a bias-free result can be achieved by choosing any $\hat\gamma < \gamma$ (Remark 3.8), though at the cost of increased noise.
> > >
> > > For the more realistic scenario without prior knowledge of $\gamma$, we introduce the underestimation and overestimation policies, both of which operate solely on the offline dataset. Due to space limits, Section 3.3 focuses on the intuitions behind these policies and the situations in which each is appropriate since we believe these topics are most relevant and crucial in practice, while the formal theoretical guarantees are deferred to Appendix C.2. These guarantees confirm the intuition: the underestimation policy restricts the inclusion of heterogeneous neighbors, while the overestimation policy ensures a sufficiently large $\hat\gamma$ to reduce noise. Our empirical results in Section 5 and Appendix F further support these findings. We emphasize that in the pure offline setting, where each user has limited data, a carefully chosen threshold is crucial for conservatively connecting edges and avoiding severe bias from heterogeneous users. Although estimating a value close to $\gamma$ is inherently challenging in this limited-data regime since some users may have very few samples which may highly restrict the quality of selected $\hat\gamma$, our proposed policies still provide reliable and robust estimates, as validated both theoretically and empirically. We consider this an important contribution of our work.
> > >
> > > Finally, Section 3.3 not only introduces several practical strategies for selecting $\hat\gamma$, but also helps validate the noise-bias tradeoff characterized in Section 3.2 and illustrated in Figure 1. Selecting a small $\hat\gamma$ (e.g., $\hat\gamma < \gamma$ or via the underestimation policy) yields lower bias but higher noise, while selecting a larger $\hat\gamma$ (e.g., via the overestimation policy) has the opposite effect. Together with the practical scenarios discussed in Remark 3.9, these results offer both theoretical and application-level insight into how $\hat\gamma$ should be chosen.
> > >
> > >
> > > **(iv) Action regularity assumption.** Due to space limitations, we refer the reviewer to W3 in our response to reviewer R7PR and W2 in our response to reviewer 1WAV for detailed discussions on why we adopt the regularity assumption and how our algorithm and analysis can be adapted to cases where this assumption does not hold.
> > >
> > > **(v) Offline bandit baselines.** Our experiments in Section 5 and Appendix F include several offline bandit baselines. In particular, LinUCB\_IND serves as an offline linear bandit method that only uses data from the test user. CLUB and SCLUB adapt traditional online clustering of bandits algorithms to the offline setting by performing one-shot decisions. In addition, XMeans\_Improve, DBSCAN\_Improve, and ARMUL represent traditional offline statistical learning baselines. Appendix F.2 provides detailed descriptions of all baselines.
> > >
> > >
> > >
> > > [1] Dai et al. (2024), Conversational recommendation with online learning and clustering on misspecified users.
> > >
> > > [2] Li et al. (2025), Demystifying online clustering of bandits: Enhanced exploration under stochastic and smoothed adversarial contexts.
> > >
> > > [3] Li et al. (2019), Improved algorithm on online clustering of bandits.
> > >
> > > [4] Wang et al. (2025), Online clustering of dueling bandits.
> > >
> > > [5] Hong et al. (2020), Latent bandits revisited.
> > >
> > > [6] Shi et al. (2023), Provably Efficient Offline Reinforcement Learning with Perturbed Data Sources.
> > >
> > > [7] Kwon et al. (2021), RL for latent mdps: Regret guarantees and a lower bound.
> > >
> > > [8] Kausik et al. (2023), Learning mixtures of Markov chains and MDPs.
> > >
> > > [9] Kausik et al. (2024), Offline Policy Evaluation and Optimization under Confounding.
> > >
> > > [10] Kausik et al. (2025), Leveraging Offline Data in Linear Latent Contextual Bandits.
> > >
> > > [11] Yang et al. (2021), Impact of representation learning in linear bandits.
> > >
> > > [12] Hu et al. (2021) Near-optimal representation learning for linear bandits and linear RL.
> > >
> > > [13] Yang et al. (2022) Nearly minimax algorithms for linear bandits with shared representation.

---

### Official Review · Reviewer_1WaV · 2025-11-06

**Soundness:** 3
**Presentation:** 3
**Contribution:** 2
**Rating:** 4
**Confidence:** 3

**Summary:**

The paper studies an offline, single-shot decision problem for contextual linear bandits under user heterogeneity: given historical logs across many users, first cluster users and then choose one action for a fresh test user to minimize the suboptimality gap. Two algorithms are proposed: Off-C2LUB (build edges among close users, aggregate one-hop neighbors, pessimistic decision) and Off-CLUB (start from a complete graph and remove edges, better when each user has sufficient data). Experiments on synthetic data and two public datasets suggest Off-C2LUB is robust under sparse logs while Off-CLUB improves with abundant data.

**Strengths:**

1: The ''cluster from logs → act once'' framing is well-motivated for applications where online interaction is limited.

2: The empty-graph vs complete-graph constructions align with low-data vs high-data regimes and are easy to implement.

3: The analysis surfaces a bias–variance trade-off around the clustering threshold and identifies a data-sufficiency regime where the complete-graph pruning approach performs well.

4: Results across synthetic and real data broadly match the narrative.

**Weaknesses:**

1: Limited technical novelty. The methods largely assemble standard components—ridge regression with confidence sets, distance-threshold user graphs, and pessimistic action selection. The paper’s contribution is more in problem formulation and tidy integration than in new algorithmic primitives or estimation techniques. The one-hop aggregation choice (vs. full component) is interesting but not theoretically pinned down as a strict improvement beyond intuition and ablations.

2: Strong regularity assumptions on actions. Several results assume known or well-behaved action covariances (or equivalent surrogates). In real logs, action distributions are policy-induced, user-dependent, and unknown. The paper does not provide learnable/estimable versions with concentration guarantees, leaving a gap between assumptions and practice.

3: Hyperparameter tuning and validation leakage. The threshold selection policies are central, yet the tuning protocol (grids, validation splits, early stopping, cross-dataset reuse) is under-specified. It’s unclear whether any distributional information from the evaluation setup leaks into selection, and sensitivity to dimension and cluster separation is not systematically reported.

4: Offline–online mismatch not considered. The paper assumes the test-time environment matches the logged data (action/context coverage, reward noise/scale), but provides no diagnostics or safeguards against distribution shift (covariate, reward, or action-set geometry) and no propensity-aware debiasing (IPS/DR) to mitigate policy-induced bias. This risks extrapolation beyond the support of the logs.

**Questions:**

See weaknesses.

---

> ### Author Response · Authors · 2025-11-21
>
> We thank the reviewer for the feedback and address the concerns below.
>
> **W1: Paper novelty.**
> We would like to clarify that, in addition to introducing a new *problem formulation*, our paper makes substantial contributions in **algorithm design** and **analysis techniques**, providing insights that go beyond prior works.
>
> First, regarding the problem formulation, our work is the first to study the offline clustering of bandits problem, extending beyond the existing literature on online clustering of bandits. The offline setting introduces a fundamental limited-data challenge, which directly breaks existing algorithmic protocols and theoretical analyses designed for online settings. This motivates the need for new methodological and analytical tools.
>
> To address this challenge, our algorithm (Off-C$^2$LUB) adopts a **novel connecting-edge policy**, which starts from a null graph and iteratively adds edges between users with confidently similar preference vectors. This stands in contrast to the traditional **deleting-edge policy** used in online works, which starts from a complete graph and removes edges when differences are detected. Our approach is intentionally more conservative and better suited to limited offline data, as it avoids aggregating data from users whose similarity cannot be reliably confirmed. This design explicitly incorporates the **cluster threshold** $\hat{\gamma}$ to control which users may be aggregated, reflecting our core principle of restricting bias under limited data.
>
> Unlike prior online works, which assume near perfect estimation of user preferences with nearly infinite online samples and therefore ignore heterogeneous bias, the offline setting requires addressing such bias carefully. Our theoretical analysis provides new technical tools by **decomposing suboptimality into noise and bias components**. In particular, Lemma 3.2 estimates the sizes of homogeneous and heterogeneous neighbor sets, while Theorem 3.4 quantifies both noise from limited aggregated samples and bias from heterogeneous neighbors. These tools are, to our knowledge, the first in this line of literature to formally capture heterogeneous bias in clustering of bandits algorithms. Furthermore, our analysis of different choices of $\hat{\gamma}$ in Section 3.3 highlights the fundamental **noise-bias tradeoff**, and provides practical strategies for different data regimes. We also analyze the traditional deleting-edge policy under offline data, showing that it performs well only when each user has sufficient samples, further reinforcing the necessity of our connecting-edge approach.
>
> Regarding the reviewer’s question about the one-hop aggregation rule, we highlight its theoretical necessity. In a word, **aggregating over entire connected components would not allow for controlled bias.** For example, consider users $u$, $v$, and $w$ where $(u,v)$ and $(v,w)$ are connected in $\mathcal{G}_{\hat\gamma}$ but $(u,w)$ is not. Then Lemma 3.2 ensures that $\|\theta_u - \theta_v\|_2$ and $\|\theta_v - \theta_w\|_2$ are each at most $\hat{\gamma}$, but the gap between $\theta_u$ and $\theta_w$ may be as large as $2\hat{\gamma}$. This example shows that aggregating over the full component would therefore violate the $\hat\gamma$ bounded-bias guarantee for heterogeneous neighbors in Lemma 3.2 and break the upper bound established in Theorem 3.4, leading to serious unrestricted bias. For this reason, one-hop aggregation is essential in the offline limited-data setting. We also discuss this choice in Appendix G when introducing the one-hop neighbor aggregation rule for the second algorithm and note that several recent online clustering works have also adopted and discussed this approach (e.g., Section IV.A in [1] and Section 3.4 in [2]).

---

> > ### Author Response · Authors · 2025-11-21
> >
> > **W2: Regularity assumption.**
> > We would like to clarify that the regularity assumption adopted in our paper (Assumption 2.1) is standard and appears in **all** prior online clustering of bandits works [1-8]. This assumption is essential in this line of research because it ensures that each user’s preference vector can be reliably estimated given sufficient data, which is necessary for correctly identifying cluster property. In our setting, we assume only that the **candidate set** $\mathcal{S}_u^i$ for each action is drawn from a distribution $\rho$ whose covariance matrix has a lower-bounded minimum eigenvalue $\lambda_a$; the **actual offline action selected** from each candidate set may be arbitrary and does not need to follow any known distribution.
> >
> > In addition, our paper provides detailed discussions of alternative versions of Assumption 2.1 as well as even scenarios where no regularity assumption holds in Remark 2.2 and Appendix B. In particular, when no regularity guarantee exists, one can modify the confidence interval $\text{CI}\_u$ in Eq. (1) of Algorithm 1 to depend directly on the minimum eigenvalue of the information matrix, rather than the number of samples and the regularity parameter. Specifically, replacing Eq. (1) with
> > $$\text{CI}\_u=\frac{\sqrt{d\log\left(1+\frac{N_u}{\lambda d}\right)+2\log\left(\frac{2U}{\delta}\right)}+\sqrt{\lambda}}{\sqrt{\lambda_{\min}(M_u)}}$$
> > removes the need for the regularity parameter entirely. The theoretical guarantee then changes accordingly, with Theorem 3.4 replaced by
> > $$\text{SubOpt}\leq \tilde O\left(
> > \sqrt{\frac{d}{\lambda\_{\min}(\mathcal{V}\_{\hat\gamma}(u\_{\text{test}}))}}
> > +\frac{\lambda\_{\max}(\mathcal{W}\_{\hat\gamma}(u\_{\text{test}}))}{\lambda\_{\min}(\mathcal{V}\_{\hat\gamma}(u\_{\text{test}}))}\cdot\hat\gamma
> > \right),$$
> > which follows directly from Eq. (11) in the appendix.
> > This alternative bound depends on the minimum eigenvalue of the aggregated information matrix, rather than the number of samples per user as in Theorem 3.4, and thus completely removes the need for Assumption 2.1 in both the algorithm and the analysis.
> >
> > We choose to retain the regularity assumption in the main paper to remain consistent with prior online clustering works and to present a more interpretable and concrete suboptimality bound. Nevertheless, our algorithm and analysis naturally extend to settings without this assumption with only minor modifications. We refer the reviewer to Appendix B for further discussion.
> >
> >
> > **W3: Experimental lacking.** We clarify that there is **no data leakage** in our parameter selection, and our tuning protocol is rigorous:
> >
> > (i) Selection of adaptive threshold ($\hat{\gamma}$): The threshold $\hat{\gamma}$ is not a hyperparameter tuned via grid search or validation splits. Instead, it is adaptively calculated solely based on training data statistics (specifically, the confidence intervals and empirical estimates) using our proposed Underestimation and Overestimation policies (defined in Eq. (5)). Since it is derived analytically from the dataset without referencing the test set, there is no leakage. We clarify that the only parameter we manually swept was $\hat{\gamma}$ (as shown in Figure 3), and this was done solely to serve as a benchmark for evaluating the effectiveness of our adaptive prediction policies against the optimal fixed value.
> >
> > (ii) Fixed hyperparameters: As detailed in Appendix F.1, the other parameters ($\lambda, \alpha, \delta$) were set to fixed constants ($\lambda=0.5, \alpha=0.1, \delta=0.01$) across all 10 random runs and all datasets. We did not perform dataset-specific tuning, ensuring a fair evaluation.
> >
> > (iii) Sensitivity analysis: regarding sensitivity to dimension ($d$) and cluster separation ($\gamma$), our theoretical analysis (Theorem 3.4 and Theorem 4.2) explicitly quantifies the dependency of the suboptimality gap on these factors (scaling with $\sqrt{d}$ and $1/\gamma$, respectively). Empirically, Figure 3 already demonstrates the robustness of our method to variations in the estimated $\hat{\gamma}$, showing that our adaptive policies consistently achieve near-optimal performance.

---

> > > ### Author Response · Authors · 2025-11-21
> > >
> > > **W4: Offline-online shift.** Regarding potential distribution shift between offline and online regimes, prior clustering of bandits works operate in fully online settings, whereas our paper specifically targets the offline case. We clarify that our work focuses on the pure offline setting with one-shot decision-making, where the test time corresponds to a single round with an arbitrary test action set $\mathcal{A}_{\text{test}}$. Thus, there is no distribution shift between the test-time and offline data: otherwise, there would be no way to learn the one-shot test data distribution with the biased offline data distribution. We can, however, offer some intuition for the hybrid offline-online scenario when the online phase has multiple rounds of online interactions. In such a setting, the learner would receive multiple additional online samples, allowing the algorithm to focus on users whose preference vectors are not yet estimated accurately. If these online samples are drawn from a different distribution than the offline data, they may provide additional coverage along underrepresented dimensions, enabling more precise estimation of user preferences and more reliable cluster identification. While the hybrid case (potentially with distribution shift) is an interesting direction, it lies outside the scope of our current work; our focus is on the offline problem and its inherent limited-data challenge. We view extending our framework to the hybrid setting as a valuable avenue for future research.
> > >
> > >
> > > [1] Dai et al. (2024), Conversational recommendation with online learning and clustering on misspecified users.
> > >
> > > [2] Li et al. (2025), Demystifying online clustering of bandits: Enhanced exploration under stochastic and smoothed adversarial contexts.
> > >
> > > [3] Gentile et al. (2014), Online clustering of bandits.
> > >
> > > [4] Li et al. (2018), Online clustering of contextual cascading bandits.
> > >
> > > [5] Li et al. (2019), Improved algorithm on online clustering of bandits.
> > >
> > > [6] Liu et al. (2022), Federated online clustering of bandits.
> > >
> > > [7] Wang et al. (2023), Online clustering of bandits with misspecified user models.
> > >
> > > [8] Wang et al. (2025), Online clustering of dueling bandits.

---

> > > > ### Comment · Reviewer_1WaV · 2025-11-25
> > > >
> > > > Thank you for the clarification on W3 and W4. I am satisfied with these answers.

---

> > ### Comment · Reviewer_1WaV · 2025-11-25
> >
> > Thank you for the clarification on W1. I now better understand what you regard as the main sources of novelty. That said, the problem studied in the paper is purely offline: all data are logged in advance and the learner makes a one-shot decision for a user already in the dataset, so the setting feels closer to clustered offline contextual learning than to an interactive problem with online exploration. The “connecting-edge” construction and the noise–bias decomposition are reasonable and technically sound, but the analysis mainly relies on standard concentration bounds and confidence intervals for linear models. Overall, I find the methodological advances useful but rather incremental.

---

> ### Comment · Reviewer_1WaV · 2025-11-25
>
> Thank you for the clarification on W2.Personally I am still not convinced by Assumption 2.1. In this purely offline, one-shot setting, the real difficulty should be that each user has very little and highly biased data. Assumption 2.1 instead assumes that, for every user, the candidate contexts come from a well-spread distribution with a uniformly bounded covariance, so that $\theta_u$ can in principle be estimated accurately given enough samples. In practice, logged contexts are driven by the logging policy and the user’s own preferences, and are unlikely to uniformly cover the feature space, so this assumption feels quite unrealistic to me. Once such a strong regularity is imposed, the analysis mostly reduces to standard concentration arguments for linear models, and I do not find the treatment of data sparsity particularly surprising from a technical point of view.

---

> ### Author Response · Authors · 2025-11-27
>
> We thank the reviewer again for the thoughtful follow-up and for acknowledging the clarifications we provided for W1, W3, and W4. We are glad that our responses addressed many of the reviewer’s concerns. Below, we respond to the remaining points in W1 and W2.
>
> **On the novelty of our analysis (W1).**
> While our analysis uses concentration bounds and confidence intervals which are indeed standard tools in linear bandits, we would like to clarify that the **core technical novelty does not lie in the tools themselves**, but rather in how they are used to analyze the offline clustering setting, where data heterogeneity and data scarcity fundamentally alter the difficulty of the problem. In contrast to prior online works which only needed to handle homogeneous neighbors and assumed essentially nearly unlimited online data, our analysis is the **first to explicitly model and handle heterogeneous neighbors when user data are fixed and limited**. This introduces several novelties with our proposed algorithms:
>
> (i) *Characterizing homogeneous vs. heterogeneous neighbors (Lemma 3.2).*
> Unlike online works where the learner can always learn the correct cluster property with nearly unlimited data, the offline learner must decide whether the connected neighbors are homogeneous or heterogeneous with the test user when each user has very few samples. Lemma 3.2 provides separate conditions for these two sets, establishing how limited data cause uncertainty and how this uncertainty propagates through the clustering procedure.
>
> (ii) *Deriving a suboptimality bound that splits into noise and bias (Theorem 3.4).*
> Prior online analyses cannot handle heterogeneous bias because online algorithms eventually separate clusters correctly with enough exploration, while in the offline setting it is impossible to learn all the homogeneous users correctly due to the limited data. By contrast, our bound in Theorem 3.4 is the first in this line of works to decompose the error into: a noise term caused by limited aggregated samples, and a bias term caused by mixing with heterogeneous users. This decomposition is central to understanding offline clustering and can motivate future works on a detailed analysis with more general settings.
>
> (iii) *Analyzing the influence of the cluster threshold $\hat{\gamma}$ (Section 3.3 \& Appendix C.2).*
> We provide two principled policies under $\gamma$ unknown case as well as analysis of the more detailed result under the case when $\gamma$ is known. We further prove their effects, and connect them to the noise–bias tradeoff. This analysis is unique to the offline setting and has no analog in prior online works.
>
> Taken together, these components form a new analytical framework for handling uncertain cluster structures under limited logged and biased data. While the concentration tools are classical, the questions we answer and the structures we analyze are new, and we believe these techniques will be valuable and inspiring for future studies of heterogeneous and data-limited regimes.

---

> > ### Author Response · Authors · 2025-11-27
> >
> > **On the regularity assumption (W2).**
> > We appreciate the reviewer’s perspective and understand the concern about realism. We would like to clarify two points:
> >
> > (i) *Why we adopt the regularity assumption in the main results.*
> > Our goal is to remain aligned with all prior online clustering of bandits works ([1–8] in references of previous replies), which rely on a regularity assumption to ensure identifiability of preference vectors. The assumption in our work is intentionally weaker. Firstly, it is placed on the candidate sets rather on the logged actions themselves. Logged actions may still concentrate to optimal actions and need not be drawn from any distribution. Furthermore, this setup also matches many real-world logged datasets like data from those online clustering of bandits policies in prior works.
> >
> > (ii) *Our analysis also applies without the regularity assumption.*
> > As detailed in Remark 2.2, Appendix B and our previous reply, by modifying the confidence interval to depend on the minimum eigenvalue of the information matrix, our algorithms remain correct and we obtain a fully valid suboptimality bound without requiring regularity assumption at all. The resulting bound depends on $\lambda_{\min}(M_u)$, which is unavoidable in the absence of structural assumptions. We chose to present the current version with the regularity assumption in the main body because it yields cleaner and more interpretable bounds in terms of sample size, and can keep our framework comparable to the whole clustering of bandits literature.
> >
> > We hope these clarifications address the reviewer’s concerns. While we agree that a less restrictive assumption can provide a more general model, we believe our setting represents a reasonable and widely adopted compromise, and our analysis meaningfully extends beyond standard concentration arguments by addressing data heterogeneity, biased logs, and limited offline samples, which are challenges not tackled in prior works.
> >
> > We thank the reviewer again for the detailed engagement and would sincerely appreciate reconsideration of the score. We are also happy to continue the discussion or provide additional clarifications if needed.

---

### Official Review · Reviewer_R7Pr · 2025-11-11

**Soundness:** 3
**Presentation:** 3
**Contribution:** 2
**Rating:** 4
**Confidence:** 4

**Summary:**

The paper considers a offline clustered linear bandit data , when there is unknown bandit parameter vector in $d$ dimensions per user and arms are sampled from a distribution whose covariance matrix has a non trivial lowest eigenvalue $\lambda_a$. There are $U$ different users. Further, the unknown bandit parameter vectors across users are clustered, that is there are effectively only $J$ distinct parameter vectors but the membership of each cluster is unknown. There $N_u$ data points per user $u$ of rewards which is a linear dot product between unknown parameter vector for that user and the arm vector pulled with additive SubGaussian noise.

The question is when a test user from this comes at test time, and a set of arms are given to you to chose from, what is the best arm you would chose depending on your estimate for the parameter vector. Clearly, offline data per user is limited and therefore one needs to accumulate data across users in the same cluster. But there may not be enough data to cluster them accurately.

Paper take the approach building a graph from an empty graph where an edge is put if the individual parameter vector estimate + confidence interval from linear gaussian concentration results over lap with another user's. There is a parameter $\hat{\gamma}$ that parameterizes the overlap or nearness of these intervals. The consideration is simple pariwise. For the final decision, the authors only look at 1-hop neighbors to accumulate all data and create a new estimate assuming all those come from the same parameter vector.

Then based on this and some confidence estimate, the best arm is chosen. Main key ideas is that authors are careful not to introduce bias by considering heterogenous users in the same cluster causing estimate bias which is hard to overcome. The parameter $\hat{\gamma}$, fact of building up from the empty graph and using only one hop neighbors keep the algorithm conservative on the bias side.

Very elaborate results as a function of $\hat{\gamma}$ is derived.

**Strengths:**

1) The paper certainly considers the offline version of the clustered linear bandit setup.

2) Paper is wary of the bias due to fixed but limited data forcing heterogenous users into clusters causing issues in parameter estimates for the decision.  Thats is interesting and commendable.

3) Error is decomposable as a $O(1/\sqrt{\lambda_a N})$ term where $N$ is the set of homogenous users identified in the test user's cluster and a bias term that depends on the inclusion of heterogenous users.

**Weaknesses:**

1)  The whole machinery revolves around standard concentration of a linear gaussian model with sub gaussian noise under the case when data matrix has lowest eigenvalue bounded below.  Everything is a more detailed manipulation of the confidence estimates with a clustering routine that aggregates users with similar parameter estimates upto a confidence estimate. While the approach to be cautious with respect to clustering heterogenous users reflected in bounds and the approach, I am quite unclear about non trivial ideas in the paper.


2) The lower bounds appears non trivial but is satisfied when there is enough data  for all users cluster correctly with high probability. What would have been useful is to have a lower bound that reflects the bias term as well.  Here again, the nontrivial nature of this bound is not clear at all.

3) The assumption of arms being sampled such that the covariance is lower bounded by a some known minimum eigenvalue is also rather strong. Usually if the data is obtained from another online bandit instance, arms closer to the best arm will be sampled more giving not very non-trivial lower bounds on the lowest  eigenvalue of the Gram matrix of arms for a user. This assumptions hides such difficulty.


4) Related works on latent bandits with cluster structure (like https://arxiv.org/abs/2301.07040, https://arxiv.org/abs/2306.13053) are missing. While these papers explore the online case, the discussion is limited to some old works and not these recent ones.

**Questions:**

1) Is it possible to extend the lower bound to the case when at finite sample, the bias term is also reflected ?

Depending on authors responses to my questions and my concerns expressed in the weakness section, I am willing to raise my score.

---

> ### Author Response · Authors · 2025-11-21
>
> Thank you for your insightful comments. We address your comments as follows:
>
> **W1: Nontrivial ideas and novelty.** We would like to clarify and highlight the novelty and nontrivial contributions of our paper in terms of the **setting**, **methodology**, and **analysis**. In summary, our work opens up the offline setting for clustering of bandits, proposes a new algorithmic protocol particularly suitable for limited-data offline environments, and develops novel theoretical tools to characterize homogeneous and heterogeneous neighbors and decompose the suboptimality into noise and bias terms, capturing the impact of limited data in a principled way.
>
> *Novelty in Setting.*
> To the best of our knowledge, this is the first work to formally consider and formulate the offline clustering of bandits problem, while all prior clustering of bandit works [1-8] focus on the online setting. We identify the unique limited-data challenge inherent in the offline scenario, where the number of samples per user is fixed and limited. This limitation causes prior **deleting-edge** based algorithms, which constitute the primary solution approach for the online clustering of bandit problem, to fail, motivating the need for a new methodological framework. Furthermore, our work also incorporates the decision-making under offline data, which requires careful aggregation of similar users' data, accurate estimations of preferences, and reliable action selection with confidence, going beyond traditional statistical learning settings.
>
> *Novelty in Methodology.*
> To address the limited-data challenge mentioned above, we propose a **connecting-edge** policy, which is in a sense the converse of deleting-edge policies. Specifically, our algorithm (Off-C$^2$LUB, Algorithm 1) starts from a null graph (in contrast to the complete graph used in all prior online works) and iteratively connects edges between users with similar estimated preferences (rather than deleting edges between dissimilar users). Intuitively, this protocol is more conservative in aggregating data and better suited to the offline setting, where estimation uncertainty is high. While the clustering condition (Eq.(2)) relies on standard concentration bounds and confidence intervals for linear models, it crucially introduces the *cluster threshold* $\hat{\gamma}$ and the minimum sample number $N_{\min}$ to control both the similarity among connected users and the minimal sample complexity required for clustering, representing another key methodological contribution beyond prior work.
>
> *Novelty in Analysis.*
> Our theoretical analysis is also the first to explicitly account for bias in the offline setting. Previous online works only analyzed in detail cases where homogeneous users were connected, which cannot handle the heterogeneous bias that naturally arises under data scarcity. Our analysis complements this by decomposing the total error into *noise* and *bias* terms, and formally establishing their tradeoff. In particular, Lemma 3.2 estimates the sizes of the homogeneous and heterogeneous neighbor sets, while Theorem 3.4 provides a suboptimality upper bound reflecting both noise and bias. The analysis of $\hat{\gamma}$ selection strategies in Section 3.3 further illustrates how this tradeoff can be effectively managed under different regimes. For completeness, we also present the offline variant of the traditional deleting-edge policy as well as the lower bound, which jointly validate the necessity and superiority of our proposed connecting-edge framework for the offline case.

---

> > ### Author Response · Authors · 2025-11-21
> >
> > **W2: Improving the lower bound.**
> > As clarified in our response to W1, the main contributions of this paper lie in introducing the novel offline clustering of bandits setting with its inherent limited data challenge, designing Off-C$^2$LUB to conservatively aggregate user data under this constraint, establishing a suboptimality bound that cleanly decomposes into noise and bias, and analyzing the noise-bias tradeoff under different $\hat\gamma$ selection strategies. The lower bound primarily serves to validate the effectiveness of our algorithms in data-sufficient regimes. While we acknowledge there exists some gap between current upper and lower bounds, we elaborate on potential ways to improve our current lower bound.
> >
> > Intuitively, under the current setting, we do not believe there exists a lower bound that can meaningfully capture the bias term. This is because we impose **no restriction on the upper bound of distances between preference vectors across clusters**. Consequently, one can always construct extreme cases where cross-cluster gaps are arbitrarily large. Thus the optimal case should avoid using those largely heterogeneous data in case of introducing arbitrarily large bias, and only focus on utilizing those homogeneous users' data. Under these scenarios, any bound that includes a bias term can be even worse than results from some trivial baselines such as LinUCB that do not consider user similarity, and thus is no doubt loose. For this reason, the lower bound in our current setting should account only for the noise term without any bias.
> >
> > However, we believe incorporating a bias term becomes possible if one further assumes an upper bound on cross-cluster distances (e.g., all heterogeneous gaps are at most some value $\Gamma$). In this scenario, whether a bias term can appear in the lower bound depends on $\Gamma$: when $\Gamma$ is small, the optimal result represented by the lower bound can include all users' data to reduce noise, at the cost of a small $\Gamma$-dependent bias term (intuitively this can also be implemented by an algorithm that aggregates data from all users indiscriminately). This small bias term may not meaningfully degrade performance and could allow the lower bound to reflect the benefit of using more samples to reduce noise, making it reasonable to represent the optimal result as the lower bound.
> >
> > For our current setting without such a bound term $\Gamma$, our intuition for improving the lower bound is to obtain a **tighter noise term**. In Theorem 4.4, the lower bound aggregates all homogeneous users. In the offline setting, however, homogeneous users naturally split into those with sufficient data and those with insufficient data. We believe a stronger lower bound would be composed of **only homogeneous users with sufficiently many samples**, rather than treating all homogeneous users identically. Intuitively, users with insufficient data cannot be reliably identified as homogeneous with the test user for any algorithm, so we believe an optimal lower bound should exclude them in the noise term. By contrast, homogeneous users with sufficient data can always be confidently identified (like users in $\mathcal{R}_{\hat\gamma}(u)$ in Algorithm 1) and thus should exist in an optimal lower bound. As noted in Remark 4.5, when each user is guaranteed to have sufficient data, both Off-CLUB (Algorithm 2) and Off-C$^2$LUB with known $\gamma$ (Corollary 3.7) match the optimal lower bound in terms of utilized sample number, further supporting our view that the sample sufficiency of each user determines whether their samples should appear in the lower bound. Therefore, to narrow the gap between our current upper and lower bounds when some users lack sufficient data, we believe a tighter lower bound is achievable by splitting users to those with sufficient and insufficient data and only includes those with sufficient data in the lower bound. A meaningful future step on this direction is to research on what is the exact gap between determining ''data-sufficient users'' and ''data-insufficient users'' to find out **how much extent of data** of users mean they have sufficient data and should exist in the lower bound.
> >
> > Both directions discussed above, including improving the noise term under the current setting and incorporating a bias term under the setting with an upper bound term $\Gamma$ for cluster differences, represent promising avenues for future research. We would be grateful for any additional feedback from the reviewer.

---

> > > ### Author Response · Authors · 2025-11-21
> > >
> > > **W3: Regularity assumption.**
> > > We would like to clarify that the action regularity assumption (Assumption 2.1) is a standard assumption adopted throughout the online clustering of bandits literature [1-8]. This assumption is necessary in this line of work because the algorithm must learn sufficient information across different dimensions about each user's preference vector in order to correctly identify its cluster.
> > >
> > > The specific concern raised by the reviewer that actions drawn from online instances can concentrate to the optimal action is actually addressed by our current version of the assumption, which directly follows recent online works [6-8]. Under this assumption, we only require that each *candidate set* $\mathcal{S}_u^i$ for each offline action is drawn from a distribution $\rho$ with a lower bounded covariance matrix. The *selected action* itself can be arbitrary within the candidate set and does **not** need to follow any fixed distribution. Therefore, even if the chosen actions concentrate heavily on optimal directions, our algorithm and analysis still hold as long as the candidate sets are generated from $\rho$.
> > >
> > > Furthermore, we provide detailed discussion of alternative assumptions and even scenarios without any regularity assumption in Appendix B. In particular, Remark B.2 explains that by modifying the confidence interval in Eq. (1) to depend on the minimum eigenvalue of the information matrix rather than the smoothed regularity term $\tilde\lambda_a$, i.e.,
> > > $$\text{CI}\_u=\frac{\sqrt{d\log\left(1+\tfrac{N_u}{\lambda d}\right)+2\log\left(\tfrac{2U}{\delta}\right)}+\sqrt{\lambda}}{\sqrt{\lambda_{\min}(M_u)}},$$
> > > our algorithm can remove the dependence on the regularity assumption altogether. In this case, the resulting suboptimality bound (originating from Eq. (11)) becomes
> > > $$\text{SubOpt}\leq \tilde O\left(\sqrt{\frac{d}{\lambda\_{\min}(\mathcal{V}\_{\hat\gamma}(u\_{\text{test}}))}}
> > > +\frac{\lambda\_{\max}(\mathcal{W}\_{\hat\gamma}(u\_{\text{test}}))}{\lambda\_{\min}(\mathcal{V}\_{\hat\gamma}(u\_{\text{test}}))}\cdot \hat\gamma\right),$$
> > > which depends on the minimum eigenvalue of the aggregated information matrix rather than the explicit number of samples, as in Theorem 3.4. To remain consistent with prior online clustering of bandits works and to obtain a more interpretable sample-based bound, we adopt the regularity assumption in the main text. However, as shown in Appendix B, **our algorithm and analysis remain valid with minor modifications even without this assumption**. We invite the reviewer to consult Appendix B for further details.
> > >
> > >
> > > **W4: Related works in latent bandits.**
> > > We thank the reviewer for pointing this out and will incorporate the relevant references in the next revision. For a detailed discussion of the differences between our model and latent bandits formulations, as well as direct comparisons, we refer the reviewer to our reply to reviewer dbFf's first question, where we address these points comprehensively.
> > >
> > >
> > > [1] Gentile et al. (2014), Online clustering of bandits.
> > >
> > > [2] Li et al. (2018), Online clustering of contextual cascading bandits.
> > >
> > > [3] Li et al. (2019), Improved algorithm on online clustering of bandits.
> > >
> > > [4] Liu et al. (2022), Federated online clustering of
> > > bandits.
> > >
> > > [5] Wang et al. (2023), Online clustering of bandits with misspecified user models.
> > >
> > > [6] Dai et al. (2024), Conversational recommendation with online learning and clustering on misspecified users.
> > >
> > > [7] Li et al. (2025), Demystifying online clustering of bandits: Enhanced exploration under stochastic and smoothed adversarial contexts.
> > >
> > > [8] Wang et al. (2025), Online clustering of dueling bandits.

---

### Meta-Review · Area_Chair_fnJr · 2025-12-08

**Summary:**

The initial scores were very mixed (8/4/4/2), and detailed responses were given on the main concerns.  At least one of the “4” reviewers had already indicated that they felt the contributions were too incremental even after reading the response.  The other reviewers’ updated recommendations are harder to guess, but my best assessment is that rejection would have still been the final decision.  Some of the reasons are given below.

**Reviewer Concerns:**

Here are some examples of remaining concerns:

1) It appears likely that 2 or 3 of the reviewers would have still had significant reservations about the novelty and significance (one explicitly said so).

2) There are many doubts around Assumption 2.1, and I am not sure that the rebuttal is entirely satisfactory.  For example:
- A list of 8 papers is given using a similar assumption, but those 8 papers appear to come from a single clique of researchers, so this does not quite point to being ubiquitous/widespread.  (Though granted, maybe these researchers have been among the main contributors to the topic.)
- The authors suggest that Appendix B removes the assumption, but the bounds there also depend on $\lambda_{\min}$, so I am doubtful of how far this goes in addressing the concern.
- The authors seem to suggest that the $\lambda_{\min}$ dependence is unavoidable, but I don’t believe this is backed up.  I could imagine that **every cluster** must have sufficient coverage, but not always **every user**.  For example, if the first entry of $\theta$ is 0 for all clusters except one, then actions of the form $(1,0,…,0)$ alone should suffice to place a user in that cluster.  This is an extreme example, but highlights a broader point raised by Reviewer 1WaV.

3) Reviewer dbFf indicated quite significant concerns on missing lines of work in the literature review, and I expect that they would at least view the required updates to the paper as non-minor.

4) I am doubtful of whether the responses would fully alleviate some other concerns, like parameter knowledge and assumptions in the upper bound, and the lower bound not capturing bias.  (e.g., for the latter, I don’t think the response of “clusters may be very far apart” is relevant, as you can instead try to prove a “minimax” style lower bound, with the worst case being when they are relatively close).

**Reviewer Scores:**

In view of the above, I expect that most of the scores would have remained unchanged.

---

### Decision · Program_Chairs · 2026-01-26

Reject